# T-Graphormer: Using Transformers for Spatiotemporal Forecasting

## Abstract

Time series data is ubiquitous and appears in all fields of study. In multivariate time series, observations are interconnected both temporally and across components. For instance, in traffic flow analysis, traffic speeds at different intersections exhibit complex spatiotemporal correlations. Modelling this dual structure poses significant challenges. Most existing forecasting methods tackle these challenges by separately learning spatial and temporal dependencies. In this work, we introduce Temporal Graphormer (T-Graphormer), a Transformer-based approach designed to model spatiotemporal correlations directly. Extending the Graphormer architecture to incorporate temporal dynamics, our method updates each node representation by selectively attending to all other nodes within a graph sequence. This design enables the model to capture rich spatiotemporal patterns with minimal reliance on predefined spacetime inductive biases. We validate the effectiveness of T-Graphormer on real-world traffic prediction benchmark datasets, achieving up to 10% reductions in both root mean squared error (RMSE) and mean absolute percentage error (MAPE) compared to state-of-the-art methods.

## 1 Introduction

Time series data is prevalent across various disciplines and appears in different forms. In retail, it manifests as customer orders over time; in finance, as stock prices; in energy grid optimization, as electricity consumption or transformer temperature (Böse et al., 2017); and in geography, as geopotential or temperature measurements. Accurate prediction of time series has long been a critical problem with significant applications, leading to the development of many techniques such as spectral analysis, linear models, and state-space models (Brockwell & Davis, 2002).

In multivariate time series, each observation at time $t$ is a vector (or matrix), indicating that the observations are not only interrelated over time but also across components. This dual structure (temporal dependence and cross-sectional dependence) presents unique challenges and opportunities for modelling. To provide an example, in a multivariate time series of economic indicators, the GDP, inflation rate, and unemployment rate are all interrelated at each time step, necessitating models that can effectively capture their complex dynamics.

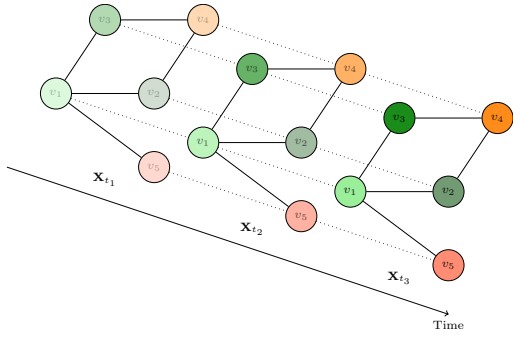

Figure 1: Visualization of a multivariate time series with graph structure. In this example, the graph remains static over time, similar to traffic networks. The node colours become more opaque to indicate more recent data points.

In this work, we focus on multivariate time series forecasting where the components are organized in a graph structure (Figure 1). Traffic flow is a prime example of such data. While traffic networks might appear "grid-like", their spatial structure is non-Euclidean. Two roads close in Euclidean space can exhibit different behaviours yet roads far in Euclidean space could exhibit similar behaviours (Li et al., 2018). For example in a city like Los Angeles, two geo-

graphically close roads (a residential street and a parallel highway) might experience vastly different traffic conditions due to differences in traffic volume, speed limits, and access points. Conversely, roads that are far in Euclidean space (two separate segments of the same highway) can exhibit similar behaviours if they are influenced by the same traffic flow dynamics. These complex spatial relationships combined with the non-stationarity of traffic flow have proven difficult for traditional machine learning techniques such as ARIMA and Kalman filtering (Okutani & Stephanedes, 1984; Lippi et al., 2013; Box et al., 2015).

Spatiotemporal forecasting, similar to advancements seen in computer vision and natural language processing (Krizhevsky et al., 2012; He et al., 2016; Radford et al., 2018; Devlin et al., 2018), has shifted from traditional machine learning methods to non-linear deep learning models. With growing efforts in data mining and improvements in computational resources, these models are increasingly capable of learning representations that capture the complex dynamics of multivariate time series, leading to more accurate predictions. Many researchers have employed convolutional modules to address spatial and temporal dependencies separately (Li et al., 2018; Yu et al., 2018; Wu et al., 2019; 2020). By interleaving temporal learning modules with spatial learning modules, these approaches facilitate the exchange of features between the time domain and the graph domain, enabling the decoupled modelling of spatiotemporal dependencies.

In this paper, we investigate whether directly modelling spatiotemporal data within a single unified framework leads to superior representation learning compared to approaches that use separate spatial and temporal learning modules. To do so, we explore variants of the Transformer architecture. Although originally designed for sequential data such as text and speech, Transformers have shown great promise in handling more complex data structures through additional encoding techniques. For instance, architectures like Graphormer (Ying et al., 2021) and Vision Transformer (ViT) (Dosovitskiy et al., 2020; Feichtenhofer et al., 2022) have successfully adapted these techniques to recover and utilize structural information when data is flattened and treated sequentially.

Building on this foundation, we introduce T-Graphormer (Temporal-Graphormer), which extends the encoding techniques of Graphormer into the temporal dimension. This approach allows T-Graphormer to leverage the global attention mechanism inherent in Transformers to capture spatial *and* temporal relationships simultaneously. Our extensive experiments on real-world traffic forecasting datasets demonstrate that T-Graphormer achieves state-of-the-art performance, surpassing existing methods by a significant margin. Furthermore, through ablation studies, we identify which encoding methods in T-Graphormer are responsible for its predictive abilities. These results highlight the potential of using Transformers as a unified framework for spatiotemporal modelling.

## 2 RELATED WORKS

In this section, we first review recent advancements in spatiotemporal forecasting. Then, we examine the relevant literature on the adoption of Transformers for various applications.

### 2.1 TRAFFIC PREDICTION

Traditional methods for time series modelling have long been foundational, with Autoregressive Moving-Average (ARMA) being a cornerstone linear model for stationary processes. To accommodate non-stationary time series, Autoregressive Integrated Moving-Average (ARIMA) was introduced (Box et al., 2015; Brockwell & Davis, 2002). Many have extended these models for multivariate time series. For instance, Lippi et al. (2013) and Williams & Hoel (2003) evaluated several statistical algorithms on traffic forecasting, including ARIMA with Kalman filters to estimate unobserved variables (the moving average components) and Support Vector Regression (SVR) models.

As deep learning showed tremendous success in Natural Language Processing (NLP) and computer vision, many borrowed ideas and architectures from these fields for traffic forecasting. Yu et al. (2018) proposed Spatio-Temporal Graph Convolutional Networks (STGCN), which uses Convolutional Neural Network (CNN) architectures to extract graphical and temporal features in traffic data. Each layer in STGCN contains a "sandwich" structure with two gated sequential convolution layers and a spatial graph convolution layer in between. Around the same time, Li et al. (2018) developed Diffusion Convolutional Recurrent Neural Network (DCRNN). Inspired by the work by Atwood & Towsley (2016), it captures spatial dependencies using diffusion convolution, which models traffic

flow as a diffusion process characterized by random walks. DCRNN then captures temporal dependencies with Recurrent Neural Network (RNN). Subsequently, Wu et al. (2019) created Graph WaveNet, which combines graph convolutional networks with the WaveNet architecture (Oord et al., 2016). In WaveNet, dilated causal convolutions (Yu & Koltun, 2016) are used to expand the receptive field for capturing historical temporal context. These convolutions are efficient, requiring fewer layers by skipping inputs based on a dilation factor that typically grows exponentially with each layer. Dilated convolutions are also utilized in Multivariate Time Series Forecasting with Graph Neural Networks (MTGNN) by Wu et al. (2020).

With the introduction of the attention mechanism in Transformers, many models began integrating it into their spatial and temporal learning modules. For instance, Attention-based Spatial-Temporal Graph Convolutional Network (ASTGCN) (Guo et al., 2019) integrates convolutional networks with temporal and spatial attention modules to selectively focus on critical information. Similarly, Zheng et al. (2020) proposed GMAN, which uses an autoencoder architecture where each block integrates spatial and temporal attention mechanisms.

More recently, Jiang et al. (2023) and Liu et al. (2023) introduced PDFormer and STAEformer, respectively. Both models utilize Transformer-based encoders for representation learning and fully connected layers for prediction. PDFormer incorporates meticulously designed self-attention modules, whereas STAEformer employs learnable spatiotemporal embeddings. Our proposed model, T-Graphormer, aligns with STAEformer's minimal reliance on architectural changes. However, T-Graphormer fundamentally differs in its approach to information processing. Unlike models with separate temporal and spatial learning modules, T-Graphormer enables all nodes to attend to one another and leverages the Transformer's inherent global attention capacity to learn coupled spatiotemporal dependencies simultaneously.

Most of the models we have discussed so far interleave temporal learning modules with spatial learning modules, transferring features between the time domain and the graph domain. In another direction, Spatial-Temporal Synchronous Graph Convolutional Network (STSGCN) simultaneously captures spatial and temporal dependencies using a novel synchronous graph convolution operation. This is done by adding positional embeddings and connecting the nodes across time steps, improving the model's ability to handle complex and evolving traffic conditions (Song et al., 2020). While this approach aligns with our goal of avoiding separate spatial and temporal learning modules, it is important to note that STSGCN treats the input as a graph and uses GCN for message-passing. In comparison, T-Graphormer processes the input as a sequence. This distinction underscores a fundamental difference in how the two models integrate spatial and temporal information.

## 2.2 TRANSFORMERS

Transformers (Vaswani et al., 2017) were originally designed for NLP applications, addressing limitations of earlier models like RNNs (Rumelhart et al., 1986) and Long Short-Term Memory (LSTM)s (Hochreiter & Schmidhuber, 1997), which struggled with long-range dependencies and training issues such as vanishing and exploding gradients (Pascanu et al., 2013). In RNN-based models, information from previous time steps is stored in a hidden state, requiring computational effort proportional to the distance between signals to relate them effectively. Transformers mitigate this challenge with an attention mechanism that processes the entire sequence as input, enabling tokens to interact directly in a constant number of operations. The multi-head attention mechanism empowers Transformers to model long-range dependencies and extract meaningful features, making them versatile for various data types, including text (Brown et al., 2020; Touvron et al., 2023; Ouyang et al., 2022), images (Dosovitskiy et al., 2020), and graphs (Ying et al., 2021; Dwivedi & Bresson, 2020; Kreuzer et al., 2021).

In NLP, Devlin et al. (2018) demonstrated that masking random tokens in sequences enables Bidirectional Encoder Representations from Transformers (BERT) to capture contextual information from both preceding and succeeding text. Since Transformers apply the attention mechanism independently to each token, they can fully leverage parallel computing. This capability allows pre-training on massive datasets, scaling to billions of parameters. As a result, pre-trained Transformers generalize effectively to downstream tasks like sentiment analysis, named entity recognition, and question answering (Radford et al., 2018; Devlin et al., 2018; Radford et al., 2019; Brown et al., 2020).

The success of Transformers in NLP inspired researchers to adapt the architecture to other data modalities. However, because Transformers are inherently designed for linear sequences, modifications are needed to accommodate non-text inputs. Dosovitskiy et al. (2020) proposed the Vision Transformer (ViT), which extends Transformers to images. ViT transforms images into a sequence of patches (or uses CNN patches directly), maps these patches to latent vectors via a trainable linear projection (patch embedding), and incorporates positional embeddings to preserve spatial structure. Unlike CNNs, ViTs have minimal image-specific inductive biases, such as convolutional kernels, while leveraging global attention mechanisms to scale efficiently.

## 3 PRELIMINARIES

This section covers the notations and definitions used in multivariate time series analysis, along with an overview of the self-attention mechanism in Transformers, which is important for understanding T-Graphormer.

### 3.1 DEFINITIONS

Formally, a time series is a set of observations $X_t$, where $t$ denotes the time of observation. While the observation across time can be continuous, we will focus on the discrete form. The forecasting of time series is the task where given $T'$ historical data at time $t$, $(X_{t-T'+1}, X_{t-T'+2}, \ldots, X_t)$, we wish to predict the next $T$ observations in the future $(X_{t+1}, X_{t+2}, \ldots, X_{t+T})$.

In multivariate time series, each observation is a vector (or a matrix). Let $X_{t,i} \in \mathbb{R}^{1 \times C}$ be the $i$th component of such observation at time $t$, and let

$$\mathbf{X}_t = (X_{t,1}^\top, X_{t,2}^\top, \ldots, X_{t,N}^\top)^\top \in \mathbb{R}^{N \times C}$$

denote the observation at time $t$ with $N$ components each having $C$ features. The goal of multivariate time series forecasting is to predict

$$\boldsymbol{\mathcal{Y}} = \left( \mathbf{X}_{t+1}^\top, \mathbf{X}_{t+2}^\top, \ldots, \mathbf{X}_{t+T}^\top \right)^\top \in \mathbb{R}^{T \times N \times C}$$

based on the historical data at time $t$

$$\boldsymbol{\mathcal{X}} = \left( \mathbf{X}_{t-T'+1}^\top, \mathbf{X}_{t-T'+2}^\top, \ldots, \mathbf{X}_t^\top \right)^\top \in \mathbb{R}^{T' \times N \times C}$$

In our traffic prediction problem, the components of an observation lie in a graph $\mathcal{G} = (\mathcal{V}, \mathcal{E}, \mathbf{W})$. Here, $\mathcal{G}$ denotes a graph with the set of vertices $\mathcal{V}$ with size $N$ and the set of edges $\mathcal{E}$ with size $M$, and $\mathbf{W} \in \mathbb{R}^{N \times N}$ denotes the weighted adjacency matrix with $\mathbf{W}_{i,j}$ being the edge length. In other words, each $i$th component of the observation $\mathbf{X}_t$ at time $t$ is the node value of node $v_i$.

### 3.2 TRANSFORMERS

The Transformer architecture is a key component in many modern deep-learning models. It consists of multiple layers, each composed of two main parts: a self-attention mechanism and a position-wise Feed-Forward Network (FFN).

Given an input sequence $H = \left( h_1^\top, \ldots, h_l^\top \right)^\top \in \mathbb{R}^{l \times d}$ where $d$ is the hidden dimension and $h_i \in \mathbb{R}^{1 \times d}$ is the hidden representation at position $i$, three matrices are used to project $H$ to obtain $Q, K, V$.

$$Q = HW_Q, \quad K = HW_K, \quad V = HW_V \tag{1}$$

where the three matrices are $W_Q \in \mathbb{R}^{d \times d_K}, W_K \in \mathbb{R}^{d \times d_K}, W_V \in \mathbb{R}^{d \times d_V}$ respectively. The similarity matrix $A$ is calculated as

$$A = \frac{QK^\top}{\sqrt{d_K}}, \tag{2}$$

which measures the semantic similarity between query vectors in $Q$ and key vectors in $K$. Finally, this attention score matrix is used to retrieve the learned value vectors:

$$\text{Attention}(H) = \text{softmax}(A)V \tag{3}$$

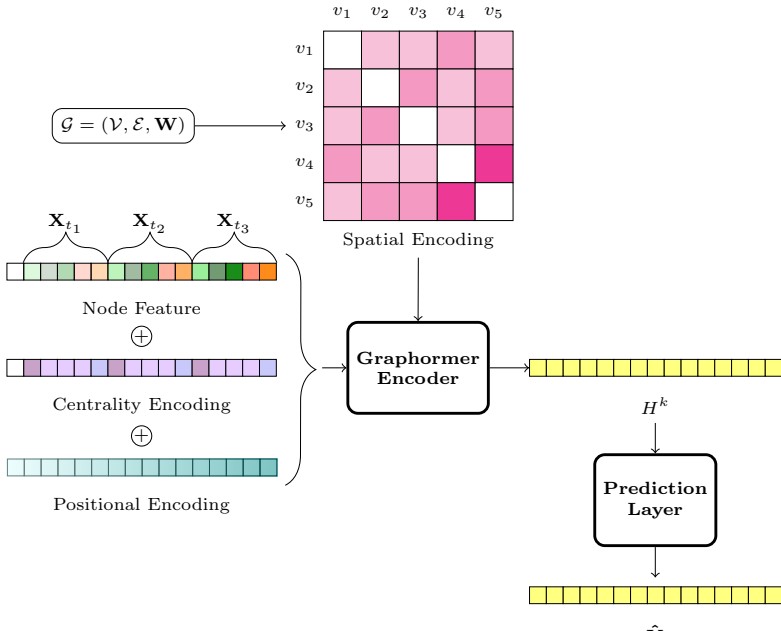

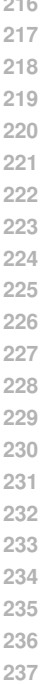

Figure 2: T-Graphormer model architecture. Graph node values across time are flattened to obtain node features. Centrality encoding and positional encoding are added to the node feature vector, which is then passed into the Graphormer encoder blocks. The edge weights of the graph are used to compute spatial encodings that determine the attention bias. Finally, the prediction layer maps the learned representation onto the output space. In this Figure, a `cls` token (white fill) is added to the beginning of the sequence. The illustration of node feature, centrality encoding, and spatial encoding vectors are consistent with the example in Figure 1.

This defines one attention head. In multi-headed attention, the outputs of each head are concatenated and projected again. After the self-attention operation, each position $i$ in the sequence $H$ is processed independently and identically by a position-wise FFN. The FFN consists of two linear transformations with an activation function in between:

$$\text{FFN}(h_i) = \text{activation}(h_i W_1 + b_1) W_2 + b_2. \tag{4}$$

## 4 T-GRAPHORMER

In this section, we discuss how Graphormer can be easily extended in the temporal dimension to produce T-Graphormer that learns from spatiotemporal data. We also provide implementation details that were beneficial in practice.

### 4.1 NODE FEATURE EMBEDDING

Following Graph WaveNet by Wu et al. (2019), we project the observed node values from $C$ to $d$ dimensions with a linear layer $W_0 \in \mathbb{R}^{C \times d}$. Let $\boldsymbol{\mathcal{X}} \in \mathbb{R}^{T' \times N \times C}$ be the historical data, we denote the initial node features as $\mathcal{X} \in \mathbb{R}^{T' \times N \times d}$.

### 4.2 STRUCTURAL ENCODINGS

As discussed in section 2, structured data must be flattened before it can be processed by Transformers. We vectorize $\mathcal{X}$ into

$$\mathbf{x} = (X_{t-T'+1,1}, \dots, X_{t+2,N}, \dots, X_{t,N}) \in \mathbb{R}^{l \times d}$$

where $l = T' \times N$. However, information is lost during this process, so when applying Transformers to spatiotemporal data, structural encoding methods are essential to introduce inductive biases that inform the model about the structure in $\mathcal{X}$. In T-Graphormer, we extend the structural encoding techniques introduced in Graphormer (Ying et al., 2021) to effectively capture the spatiotemporal relationships within the data. This is mainly done in two ways: modifications to the node features and modifications to the attention mechanism (see Figure 2).

Let $x_{t,i} \in \mathbb{R}^{1 \times d}$ be the feature vector of the $i$th node at time $t$, centrality encoding is added to $x_{t,i}$ to inform the model about node importance. Intuitively, in a traffic network, intersections with more road connections have significant effect on the downstream traffic conditions. Let $\deg^-(v)$ denote the in-degree of node $v$, and $\deg^+(v)$ denote the out-degree, centrality encoding, $Z^- \in \mathbb{R}^{\deg^-(V) \times d}$ and $Z^+ \in \mathbb{R}^{\deg^+(V) \times d}$, are real-valued learnable embedding matrices indexed by $\deg(v)$ where the number of rows corresponds to the maximum node in-degree and out-degree in $\mathcal{G}$ respectively. Concretely, centrality encoding is applied to all initial node features before entering the first Transformer block

$$h_{t,i}^0 = x_{t,i} + z_{\deg^-(v_i)}^- + z_{\deg^+(v_i)}^+. \tag{5}$$

Note that the encoding is time-agnostic and is determined solely by the degree of the corresponding node. In other words, for $x_{t_1,i}$ and $x_{t_2,i}$ the centrality encodings are both $z_{\deg}(v_i)$ (as shown in Figure 2 centrality encoding). Also in cases of undirected graphs, only $Z \in \mathbb{R}^{\deg(V) \times d}$ is used.

To account for the temporal structure in multivariate time series, we also add learnable positional encoding (Vaswani et al., 2017; Dosovitskiy et al., 2020) to the initial node features. Let $P \in \mathbb{R}^{l \times d}$ denote the learned matrix where $l = T'N$, we update the initial node features in equation 5 as

$$h_{t,i}^0 = x_{t,i} + z_{\deg^-(v_i)}^- + z_{\deg^+(v_i)} + p_{t,i}. \tag{6}$$

Note that, unlike centrality encoding, positional encoding vectors $p_{t,i}$ are specific to each token in the vectorized time series. While Feichtenhofer et al. (2022) adopts separate positional embeddings when applying ViT on videos, with one for time and one for space, we find T-Graphormer performs better when the spatiotemporal positional embedding is learned simultaneously.

While the attention mechanism in Transformers has an effective global receptive field, the spatial encoding method is another important structural bias that improves spatial learning. Given the graph network $\mathcal{G} = (\mathcal{V}, \mathcal{E}, \mathbf{W})$, the model can leverage the weighted adjacency matrix to determine which nodes are closer in topological space for updating representations. Let $\phi(v_i, v_j) : \mathcal{V} \times \mathcal{V} \mapsto \mathbb{R}$ denote a function that measures the spatial relation between nodes, the attention score (equation 2) between tokens $(t_1, i)$ and $(t_2, j)$ is updated as:

$$A_{(t_1,i),(t_2,j)} = \frac{(h_{t_1,i} W_Q)(h_{t_2,i} W_K)^\top}{\sqrt{d}} + b_{\phi(i,j)} \tag{7}$$

where $b_{\phi(i,j)}$ is the learnable attention bias scalar indexed by $\phi(i, j)$ from $\Phi$.

Similar to Graphormer, we define $\phi$ as the shortest path distance (SPD) between nodes and $\Phi \in \mathbb{R}^{N \times N}$ as the matrix storing the SPD values for all node pairs. For example, if the longest path between any two nodes in $\mathcal{G}$ is 3, the learnable attention bias embedding $B$ will have shape $3 \times 3 \times$ *number of heads*. Like centrality encoding, spatial encoding is time-independent; for nodes $i$ and $j$, the term remains consistent across time steps (see Figure 2, Spatial Encoding).

### 4.3 IMPLEMENTATION DETAILS

To project the learned node representations from the encoder to the output space, we explore two architectures. In the first, a vanilla setup, we use two linear layers to project the hidden representation at the last layer $H_k \in \mathbb{R}^{l \times d}$ from $d$ dimensions to $\frac{d}{2}$ and then to $C$. In the second setup, we introduce additional dilated causal convolutional layers (Yu & Koltun, 2016). This imposes a time bias that ensures future predictions are based solely on representations from previous time steps. The output from these convolutional layers is then projected to $C$ dimensions using the same linear layers.

Before passing the node features through the Transformer blocks, we experiment with adding special tokens to the sequence:

- No special token added, the length of the sequence is $T' \times N$ before passing through the encoder.

- Adding a `cls` token to the start of the sequence (see Figure 2), resulting in $(T' \times N) + 1$ tokens. The implementation is similar to the virtual node in Graphormer, where a learnable embedding of size $\mathbb{R}^{1 \times d}$ is introduced and concatenated with the initial node features $\mathcal{X}$. This token acts as a supernode, aggregating information from the entire sequence and propagating it back to other tokens (Ying et al., 2021).

- Adding a `graph` token to the start of each graph signal, resulting in $T' \times (N + 1)$ tokens. The implementation is similar to the `cls` token, with a learnable embedding used to represent the `graph` token. Its function is analogous to the `sep` token in BERT (Devlin et al., 2018), serving as a delimiter for graph signals at different time steps.

For all special tokens, we form virtual connections to other nodes. $b_{\phi(\texttt{token},v)}$ is learned separately.

Following what has been observed with current Transformer implementations, we find that using layer normalization (Ba et al., 2016) before multi-headed attention and FFN improves T-Graphormer performance. We also find Gaussian Error Linear Unit (GELU) to be the best activation function.

## 5 EXPERIMENTS

In this section, we detail the experimental settings used to evaluate T-Graphormer on two traffic prediction datasets. We compare T-Graphormer's performance against 10 baseline methods. Additionally, to gain deeper insights into T-Graphormer, we conduct ablation studies focusing on its structural encodings and the impact of added special tokens.

### 5.1 DATASETS

Following Li et al. (2018); Wu et al. (2019; 2020); Shao et al. (2022), we focus on evaluating T-Graphormer on two commonly used spaiotemporal forecasting datasets (see Table 1 for details):

- PEMS-BAY (Chen et al., 2001): A traffic speed dataset collected from California Transportation Agencies (CalTrans) Performance Measurement System (PeMS).

- METR-LA (Jagadish et al., 2014): A traffic speed dataset collected from loop detectors on the highway of Los Angeles County.

| Datasets | # Sensors | # Samples | Sampling Rate | Mean | Std |
|---|---|---|---|---|---|
| PEMS-BAY | 325 | 52116 | 5 minute | 54.40 | 19.49 |
| METR-LA | 207 | 34727 | 5 minute | 62.73 | 9.44 |

Table 1: Details of the dataset used for evaluation. The mean and standard deviation (std) are calculated across time and space.

For both datasets, we follow the same pre-processing implementation as Li et al. (2018)[1]. Specifically, each sample $\mathbf{X}_t$ is a 5-minute traffic speed reading from all sensors in the network. We aggregate 12 consecutive samples (representing a 1-hour context window) to construct $\mathcal{X} \in \mathbb{R}^{12 \times N \times 1}$. The ground truth $\mathcal{Y}$ consists of the next $T$ future traffic speed readings from all sensors. To ensure consistency with the baseline methods, we evaluate the models on predictions for $T = \{3, 6, 12\}$, corresponding to 15 minutes, 30 minutes, and 1 hour, respectively. We also concatenate a one-hot-encoding vector signifying the time of day to all node values at time $t$.

The data is split such that approximately 70% is used for training, 20% for testing, and 10% for validation. To avoid data leakage in the traffic prediction task, the time slices are kept in their original order before splitting. Only the training dataset is shuffled for different training iterations and used to perform $Z$-score normalization on the validation and testing datasets. To construct the graph $\mathcal{G} = (\mathcal{V}, \mathcal{E}, \mathbf{W})$, we follow the same implementation as Li et al. (2018) and refer readers to that work for further details.

---

[1]https://github.com/liyaguang/DCRNN

| Dataset | Method | Horizon 3 | | | Horizon 6 | | | Horizon 12 | | |
|---------|--------|-----|------|----------|-----|------|------|-----|------|------|
| | | MAE | RMSE | MAPE (%) | MAE | RMSE | MAPE | MAE | RMSE | MAPE |
| **PEMS-BAY** | VAR | 1.74 | 3.16 | 3.60 | 2.32 | 4.25 | 5.00 | 2.93 | 5.44 | 6.50 |
| | FC-LSTM | 2.05 | 4.19 | 4.80 | 2.20 | 4.55 | 5.20 | 2.37 | 4.96 | 5.70 |
| | DCRNN | 1.38 | 2.95 | 2.90 | 1.74 | 3.97 | 3.90 | 2.07 | 4.74 | 4.90 |
| | STGCN | 1.36 | 2.96 | 2.90 | 1.81 | 4.27 | 4.17 | 2.49 | 5.69 | 5.79 |
| | Graph WaveNet | 1.30 | 2.74 | 2.73 | 1.63 | 3.70 | 3.67 | 1.95 | 4.54 | 4.63 |
| | ASTGCN | 1.52 | 3.13 | 3.22 | 2.01 | 4.27 | 4.48 | 2.61 | 5.42 | 6.00 |
| | STSGCN | 1.44 | 3.01 | 3.04 | 1.83 | 4.18 | 4.17 | 2.26 | 5.21 | 5.40 |
| | GMAN | 1.34 | 2.91 | 2.86 | 1.63 | 3.76 | 3.68 | 1.86 | 4.32 | 4.37 |
| | PDFormer | 1.32 | 2.83 | 2.78 | 1.64 | 3.79 | 3.71 | 1.91 | 4.43 | 4.51 |
| | STAEformer | 1.31 | 2.78 | 2.76 | 1.62 | 3.76 | 3.62 | 1.88 | 4.34 | 4.41 |
| | STEP | **1.26** | 2.73 | **2.59** | 1.55 | 3.58 | 3.43 | 1.79 | 4.20 | 4.18 |
| | T-Graphormer | 1.31 | **2.55** | 2.71 | **1.52** | **3.14** | 3.23 | **1.76** | **3.78** | **3.91** |
| **METR-LA** | VAR | 4.42 | 7.80 | 13.00 | 5.41 | 9.13 | 12.70 | 6.52 | 10.11 | 15.80 |
| | FC-LSTM | 3.44 | 6.30 | 9.60 | 3.77 | 7.23 | 10.09 | 4.37 | 8.69 | 14.00 |
| | DCRNN | 2.77 | 5.38 | 7.30 | 3.15 | 6.45 | 8.80 | 3.60 | 7.60 | 10.50 |
| | STGCN | 2.88 | 5.74 | 7.62 | 3.47 | 7.24 | 9.57 | 4.59 | 9.40 | 12.70 |
| | Graph WaveNet | 2.69 | 5.15 | 6.90 | 3.07 | 6.22 | 8.37 | 3.53 | 7.37 | 10.01 |
| | ASTGCN | 4.86 | 9.27 | 9.21 | 5.43 | 10.61 | 10.13 | 6.51 | 12.52 | 11.64 |
| | STSGCN | 3.31 | 7.62 | 8.06 | 4.13 | 9.77 | 10.29 | 5.06 | 11.66 | 12.91 |
| | GMAN | 2.80 | 5.55 | 7.41 | 3.12 | 6.49 | 8.73 | 3.44 | 7.35 | 10.07 |
| | PDFormer | 2.83 | 5.45 | 7.77 | 3.20 | 6.46 | 9.19 | 3.62 | 7.47 | 10.91 |
| | STAEformer | 2.65 | 5.11 | 6.85 | 2.97 | 6.00 | 8.13 | **3.34** | 7.02 | 9.70 |
| | STEP | **2.61** | **4.98** | **6.60** | 2.96 | **5.97** | 7.96 | 3.37 | 6.99 | 9.61 |
| | T-Graphormer | 2.67 | 5.37 | 6.65 | **2.94** | 5.98 | **7.46** | 3.35 | **6.92** | **8.61** |

Table 2: Spatiotemporal forecasting results on two traffic network datasets. We report the results from the best configuration of T-Graphormer on each dataset. The best-performing model for each metric is bolded, and the second-best-performing model is underlined. Details of the datasets and metrics are described in sections 5.1 and 5.2

.

## 5.2 SETTINGS

We conduct extensive experiments to investigate how different hyperparameters influence the performance of T-Graphormer. Specifically, we evaluate 2 different Transformer configurations with $d = \{128, 384\}$, $k = \{6, 10\}$, and number of heads $= \{4, 8\}$. These configurations are chosen to maximize the available hardware memory. As detailed in section 4.3, we test 2 distinct prediction layer architectures. For dilated causal convolutions, we select a dilation factor of 2 and use 3 layers.

Training of T-Graphormer is guided by minimizing the mean squared error (MSE) between predicted values $\hat{\mathbf{Y}}$ and ground truth $\mathcal{Y}$ for $T = 12$ only. When evaluating the prediction performance on horizons 3 and 6, we simply remove the redundant token values. We employ the AdamW optimizer (Loshchilov & Hutter, 2019) with an effective batch size of 128, using a cosine decay learning rate schedule with warmup (Loshchilov & Hutter, 2017). After training, model configurations with the lowest mean absolute error (MAE) on the validation dataset are selected for testing. T-Graphormer is implemented in `PyTorch` (Paszke et al., 2019) and utilizes distributed data parallelism to speed up training and increase batch size.

T-Graphormer is compared against the following baseline methods: 1. Vector Autoregressive (VAR) (Zivot & Wang, 2006), 2. Support Vector Regression (SVR) (Smola & Schölkopf, 2004), 3. FC-L-STM (Sutskever et al., 2014), 4. Graph Multi-Attention Network (GMAN) (Zheng et al., 2020), 5. STGNN is Enhanced by scalable time series Pre-training model (STEP) (Shao et al., 2022), 6. PDFormer (Jiang et al., 2023), 7. STAEformer (Liu et al., 2023). Additional baselines, including DCRNN, STGCN, Graph WaveNet, MTGNN, ASTGCN, and STSGCN, are discussed in section 2.1. It is worth noting that STEP leverages masked pre-training on long-term historical time series data, enabling models to capture long-term dependencies.

For more information on experimental settings, see Appendix at section A.

## 5.3 MAIN RESULTS

T-Graphormer demonstrates exceptional performance in long-range traffic prediction. As shown in Table 2, T-Graphormer outperforms all other models across every metric when predicting the

next 12 time steps (1-hour window). Notably, with minimal spacetime inductive bias, it achieves a **10.00%** reduction in RMSE on the PEMS-BAY dataset and a **10.40%** reduction in MAPE on the METR-LA dataset compared to the state-of-the-art model STEP. We attribute this success to the global receptive field provided by the self-attention mechanism, a key feature of Transformers.

However, T-Graphormer underperforms in short-range traffic prediction. When forecasting the next 15 minutes, it consistently lags behind STEP across all metrics. One possible explanation is that T-Graphormer's reliance on structural encoding methods is limiting its ability to capture local space-time patterns which are critical for short-term predictions. Another factor could be overfitting, as the model is optimized by backpropagation on the prediction error for the next hour (Horizon 12), and short-range predictions (Horizon 3 and 6) are evaluated by masking the extra tokens.

We observe that T-Graphormer performed differently across the two datasets (see Table 6) as the prediction task on METR-LA is harder. In Figure 5, we see that despite overfitting, the training loss of the best model is much lower in PEMS-BAY (9.27) than in METR-LA (55.28). The difficulty of the task is further illustrated in Figure 4 where the validation MAPE is around 4 for PEMS-BAY but increased to around 9 for METR-LA. This is unsurprising since the traffic speed standard deviation (19.49) of METR-LA is much higher than that of PEMS-BAY (9.44), meaning there is more variability in the road conditions of the Los Angeles traffic grid.

Additionally, when comparing the training and validation loss over epochs in Figure 5, the 2 loss curves follow much more closely in PEMS-BAY than in METR-LA, a sign that the model is significantly overfitting on METR-LA. From the dataset side, this can be explained by the fact that PEMS-BAY has 50% more samples than METR-LA (52116 vs. 34727), which allows the model to fully learn the spatiotemporal relationships between flattened tokens on the PEMS-BAY dataset.

This overfitting behaviour is inspected from another perspective in Figure 6. Between the 3 model sizes, although the initial loss differences appear random, a trend emerges towards the end of training, where the bigger the model, the bigger the loss difference between training and validation. This overfitting behaviour is consistent with the recent findings on the empirical scaling laws for training LLM (Kaplan et al., 2020). Using equation (6.6) from this work: $D \propto N^{0.74}$, we find that the traffic prediction dataset sizes are only optimal for training the mini models, but too small for the larger models and cause overfitting.

## 5.4 ABLATION EXPERIMENTS

In this section, we examine the efficacy of the added structural encoding methods and assess the impact of special tokens. This is done by re-training the best-performing model with the same configuration but with the missing components.

It is evident from Table 3 that positional encoding and spatial encoding are the critical structural and temporal inductive biases when applying Transformers to spatiotemporal data. When positional encoding is removed, the model fails to predict accurately, leading to a **15.22%** increase in MAE. This is unsurprising since Transformers lack recurrence or convolutional mechanisms and thus rely on positional encoding to capture temporal information.

When spatial encoding is removed, the performance drops **8.06%**. This is consistent with findings from the growing literature in traffic prediction (Li et al., 2018; Yu et al., 2018; Wu et al., 2019; Guo et al., 2019), where adding structural biases (e.g. in the form of using Graph Neural Net-

| Model | | | | Horizon 3 | | | Horizon 6 | | | Horizon 12 | | |
|---|---|---|---|---|---|---|---|---|---|---|---|---|
| Positional | Spatial | Centrality | Token | MAE | RMSE | MAPE | MAE | RMSE | MAPE | MAE | RMSE | MAPE |
| ✗ | ✓ | ✓ | ✗ | 3.29 | 6.87 | 8.55 | 3.51 | 7.42 | 9.18 | 4.12 | 9.04 | 10.53 |
| ✓ | ✗ | ✓ | ✗ | 3.35 | 6.82 | 8.45 | 3.51 | 7.23 | 8.91 | 3.89 | 8.15 | 9.83 |
| ✗ | ✓ | ✓ | cls | 3.15 | 6.34 | 8.33 | 3.38 | 6.93 | 8.94 | 3.86 | 8.14 | 10.07 |
| ✓ | ✗ | ✓ | cls | 3.07 | 6.36 | 7.60 | 3.29 | 6.99 | 8.24 | 3.62 | 7.71 | 9.22 |
| ✓ | ✓ | ✗ | cls | 2.75 | 5.52 | 7.15 | 3.05 | 6.26 | 8.02 | 3.51 | 7.31 | 9.24 |
| ✓ | ✓ | ✓ | ✗ | 2.78 | 5.73 | 7.16 | 3.11 | 6.56 | 8.03 | 3.50 | 7.38 | 9.10 |
| ✓ | ✓ | ✗ | ✗ | 2.74 | 5.64 | 7.30 | 3.05 | 6.42 | 8.19 | 3.45 | 7.35 | 9.36 |
| ✓ | ✓ | ✓ | graph | 2.81 | 5.79 | 7.47 | 3.08 | 6.41 | 8.22 | 3.42 | 7.01 | 9.22 |
| ✓ | ✓ | ✓ | cls | 2.67 | 5.37 | 6.65 | 2.94 | 5.98 | 7.46 | 3.35 | 6.92 | 8.61 |

Table 3: Ablation results on the METR-LA dataset. The rows are sorted by descending order on horizon 12 prediction MAE.

work (GNN)s) generally improves prediction performance. When positional and spatial encodings are both removed, the drop in performance we observe is additive such that MAE is increased by **22.98%**.

Removing centrality encoding has a less pronounced effect, with only a 4.78% increase in MAE. This limited impact may be attributed to the fact that in our setting, graphs are static over time, so with enough learning, positional encoding can provide sufficient information about node importance. Consistent with the work by Ying et al. (2021), we also observe that adding special tokens improves performance. We also find that adding `cls` or `graph` token noticeably improves prediction performance, and `cls` is generally better.

Interestingly, when `cls` token and centrality encoding are *both* excluded (row 7), the MAE increases to 3.45, which is lower than the error increase when either feature is excluded individually (3.51 and 3.50 in rows 5 and 6). This indicates that models perform better when the features are excluded together. We find this behaviour explainable in one direction: without node degree embedding, it is better to also exclude `cls` token. Since centrality encoding is applied only to the real nodes in the graph signal, it helps the model distinguish regular nodes from the `cls` token. Without it, the `cls` token confuses the model. However, the other direction (without `cls` token, it is better to also exclude centrality encoding) is more difficult to explain. It could be that when both features are present, they contribute complementary information to the model, but when both are excluded, the model adapts by leveraging other available features, such as positional encoding, to compensate for the missing information about node centrality.

## 6 CONCLUSION

We introduce a novel framework for modelling multivariate time series. By leveraging structural encoding methods and extending them along the temporal dimension, we show that the Transformer architecture can be directly applied to spatiotemporal data without separate temporal and spatial learning modules. This integration allows T-Graphormer to capture spatiotemporal dependencies simultaneously with minimal spacetime inductive bias. Building on this foundation, future work can readily incorporate domain knowledge into the architecture, such as using adaptive spatiotemporal embeddings from STAEformer and custom spatial attention mechanisms from PDFormer.

However, our study has certain limitations that merit discussion. While T-Graphormer exhibits strength in modelling spatiotemporal data, it incurs high memory demands due to its $\mathcal{O}(n^2)$ complexity. Flattening the entire time series into a sequence significantly increases context length as more historical data is included. For example, in the PEMS-BAY dataset, adding one additional time step increases the context length by 325. This dramatically impacts memory usage, constraining T-Graphormer's applicability to large networks or longer time windows. For instance, training T-Graphormer on the large-scale traffic forecasting dataset LargeST (Liu et al., 2024), which includes up to 8600 nodes, poses significant challenges. However, incorporating techniques such as sparse attention (Beltagy et al., 2020) could mitigate this issue.

Additionally, this work focuses on applying T-Graphormer to static graphs, where the graph structure remains constant over time. Dynamic graphs, where the graph structure evolves, present a promising area for extension. For instance, Shang et al. (2021) demonstrated that graph learning techniques can effectively re-parameterize dynamic graphs for forecasting tasks. Adapting T-Graphormer to dynamic graphs would require strategies such as introducing a maximum graph size and applying padding to handle structural changes over time. These enhancements can expand T-Graphormer's applicability to a broader range of spatiotemporal forecasting tasks and provide deeper insights into its graph representation learning capabilities. Extending T-Graphormer to other data modalities with grid-like structures presents another exciting avenue. Potential applications include video data (Han et al., 2022) and geographical measurements (Bi et al., 2023).

Transformer's success in NLP can be attributed to its self-supervised pre-training technique (Devlin et al., 2018; Radford et al., 2018; 2019). Following the approaches of Feichtenhofer et al. (2022) and Shao et al. (2022), T-Graphormer can be readily utilized as a masked autoencoder on spatiotemporal data. Our preliminary results indicate that masked pre-training can improve T-Graphormer training stability and reduce training time (see Figure 7).

# 7 REPRODUCIBILITY STATEMENT

For reproducibility, we have included links to the anonymized code repository, the datasets, and the pre-trained model weights in section A.1. We also summarize the training configurations and environments. All experimental logs are saved and available online at Weights and Biases upon request.

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

# A APPENDIX

## A.1 ADDITIONAL IMPLEMENTATION DETAILS

To construct the sensor graph, we compute the pairwise road network distances between sensors and build the adjacency matrix using thresholded Gaussian kernel (Li et al., 2018). $W_{i,j} = \exp\left(-\frac{\text{dist}(v_i, v_j)^2}{\sigma^2}\right)$ if $\text{dist}(v_i, v_j) \leq \kappa$, otherwise 0, where $W_{i,j}$ is the edge distance between nodes $i, j$, and $\text{dist}$ measures the physical distance between the nodes. $\sigma$ is the standard deviation of the distances, and $\kappa$ is the threshold for determining if an edge exists.

We also use the training dataset to $Z$-score normalize the entire dataset. In cases where the time of day information is available for the dataset, we concatenate it to the traffic speed measurement to enrich the input.

Besides the details mentioned in the main text, we also use gradient accumulation to increase batch size. This results in an effective batch size that is calculated by multiplying the original batch size by the number of GPUs (graphics processing units) used in distributed data parallelism and the number of accumulated gradient iterations. We also utilize layer-wise learning rate decay following (Raffel et al., 2020), where the learning rate decays exponentially in earlier layers. This is typically used to stabilize training when fine-tuning Transformer models. We find using gradient clipping and dropout improves T-Graphormer training as well (Srivastava et al., 2014).

We summarize our training configurations in Tables 4 and 5. All training has been done on SLURM workload manager environments. For T-Graphormer mini and small models (including causal mini and causal small), training was done on 4 compute nodes. Each compute node has 187 gigabytes of memory, 2 Intel Silver 4216 Cascade CPUs (central processing units), and 4 NVIDIA V100 Volta GPUs with 32 gigabytes of memory. For training the medium-sized models, we used 4 compute nodes with 498 gigabytes of memory, 2 AMD Milan 7413 CPUs, and 4 NVIDIA A100SXM4 GPUs with 40 gigabytes of memory. On the PEMS-BAY dataset, training validation and model checkpoint took an average of 7.5 hours for mini-sized models, 13.8 hours for small-sized models, and 10.0 hours for medium-sized models (due to GPU difference). On the METR-LA dataset, it took an average of 4.5 hours for mini-sized models, 8.5 hours for small-sized models, and 10.0 hours for medium-sized models. When training on the PEMS-BAY dataset, we also find that medium-sized models tend to overfit after 22 epochs, so we evaluate the saved models at the 22nd epoch. For per epoch training time on the METR-LA dataset, see Figure 7.

- Code: `https://anonymous.4open.science/r/t-graphormer/gmae_st/`
- Datasets (Li et al., 2018; Song et al., 2020): `https://github.com/liyaguang/DCRNN,https://github.com/Davidham3/STSGCN`
- Pre-trained model weights: `https://drive.google.com/drive/folders/1xRyxP_K0y5NoMkwzXFIgdt-6elLo3xdK?usp=drive_link`

## A.2 ADDITIONAL RESULTS

We also analyze the scalability of the models by evaluating their performances on the testing dataset. The results for the best-performing models with configurations listed in section 5.2 are shown in

| Configuration | Model | | | | | |
|---|---|---|---|---|---|---|
| | mini | small | medium | causal mini | causal small | causal medium |
| optimizer | AdamW (Loshchilov & Hutter, 2019) | | | | | |
| optimizer momentum | $\beta_1, \beta_2 = 0.9, 0.999$ | | | | | |
| learning rate schedule | cosine decay (Loshchilov & Hutter, 2017) | | | | | |
| hidden dimension ($d$) | 128 | 192 | 384 | 128 | 192 | 384 |
| epochs | 50 | 50 | 30 | 50 | 50 | 30 |
| learning rate | 1.50e-3 | 3.00e-3 | 1.25e-3 | 1.25e-3 | 1.50e-3 | 1.50e-3 |
| gradient clipping | 1.0 | 1.0 | 1.0 | 1.0 | 1.0 | 1.0 |
| weigh decay | 1e-4 | 1e-4 | 1e-4 | 1e-4 | 1e-4 | 1e-5 |
| warmup epochs | 10 | 10 | 10 | 10 | 10 | 10 |
| batch size | 128 | 96 | 96 | 128 | 96 | 96 |
| dropout | 0.1 | 0.1 | 0.1 | 0.1 | 0.1 | 0.1 |
| layer-wise decay | 0.90 | 0.75 | 0.90 | 0.90 | 0.90 | 0.90 |
| # of parameters (M) | 1.76 | 4.44 | 19.61 | 1.91 | 4.76 | 20.91 |

Table 4: Best training hyperparameters of T-Graphormer on PEMS-BAY dataset.

| Configuration | Model | | | | | |
|---|---|---|---|---|---|---|
| | mini | small | medium | causal mini | causal small | causal medium |
| optimizer | AdamW (Loshchilov & Hutter, 2019) | | | | | |
| optimizer momentum | $\beta_1, \beta_2 = 0.9, 0.999$ | | | | | |
| learning rate schedule | cosine decay (Loshchilov & Hutter, 2017) | | | | | |
| hidden dimension ($d$) | 128 | 192 | 384 | 128 | 192 | 384 |
| epochs | 100 | 100 | 100 | 100 | 100 | 30 |
| learning rate | 4.50e-3 | 1.25e-3 | 1.50e-3 | 4.85e-3 | 5.85e-3 | 1.00e-3 |
| gradient clipping | 5.0 | 5.0 | 5.0 | 5.0 | 5.0 | 5.0 |
| weigh decay | 1e-5 | 1e-5 | 1e-4 | 1e-5 | 1e-6 | 1e-6 |
| warmup epochs | 30 | 30 | 30 | 30 | 30 | 30 |
| batch size | 128 | 128 | 128 | 128 | 128 | 128 |
| dropout | 0.1 | 0.1 | 0.1 | 0.1 | 0.1 | 0.1 |
| layer-wise decay | 0.90 | 0.90 | 0.90 | 0.90 | 0.90 | 0.90 |
| # of parameters (M) | 1.57 | 4.15 | 19.04 | 1.68 | 4.48 | 20.34 |

Table 5: Best training hyperparameters of T-Graphormer on METR-LA dataset.

| Dataset | Model | Horizon 3 | | | Horizon 6 | | | Horizon 12 | | |
|---|---|---|---|---|---|---|---|---|---|---|
| | | MAE | RMSE | MAPE (%) | MAE | RMSE | MAPE | MAE | RMSE | MAPE (%) |
| PEMS-BAY | **mini** | **1.31** | **2.55** | **2.71** | **1.51** | **3.14** | **3.25** | **1.76** | **3.78** | **3.91** |
| | small | **1.31** | 2.66 | 2.77 | 1.54 | 3.33 | 3.37 | 1.79 | 3.99 | 4.04 |
| | medium | **1.31** | 2.62 | 2.76 | 1.53 | 3.22 | 3.32 | 1.77 | 3.85 | 3.99 |
| | causal mini | 1.33 | **2.55** | 2.91 | 1.58 | 3.19 | 3.56 | 1.84 | 3.84 | 4.27 |
| | causal small | 1.35 | 2.78 | 3.01 | 1.60 | 3.52 | 3.69 | 1.88 | 4.21 | 4.44 |
| | causal medium | 1.90 | 4.14 | 4.65 | 2.00 | 4.35 | 4.90 | 2.13 | 4.64 | 5.25 |
| METR-LA | mini | 2.69 | 5.76 | 6.75 | 3.01 | 6.65 | 7.60 | 3.52 | 7.92 | 8.88 |
| | small | 2.70 | 5.98 | 6.53 | 3.06 | 7.03 | 7.54 | 3.62 | 8.42 | 9.01 |
| | medium | 2.69 | 5.86 | 6.75 | 3.02 | 6.77 | 7.65 | 3.50 | 7.98 | 8.96 |
| | **causal mini** | **2.67** | 5.37 | **6.65** | **2.94** | **5.98** | **7.46** | **3.35** | **6.92** | **8.61** |
| | causal small | 2.70 | **5.30** | 6.96 | 3.01 | 6.06 | 7.91 | 3.45 | 7.06 | 9.20 |
| | causal medium | 2.91 | 6.06 | 7.19 | 3.33 | 7.11 | 78.36 | 3.96 | 8.73 | 9.98 |

Table 6: Additional spatiotemporal forecasting results on two traffic network datasets. The row with the best metric for each prediction length is bolded. The overall best-performing model for each dataset is also bolded.

| Datasets | # Sensors | # Samples | Sampling Rate | Mean | Std |
|---|---|---|---|---|---|
| PEMS03 | 358 | 26185 | 5 minute | 181.38 | 144.41 |
| PEMS04 | 307 | 16969 | 5 minute | 207.23 | 156.48 |
| PEMS08 | 170 | 17833 | 5 minute | 229.86 | 145.62 |

Table 7: Details of the additional dataset used for evaluation.

| Method | PEMS03 | | | PEMS04 | | | PEMS08 | | |
|---|---|---|---|---|---|---|---|---|---|
| | MAE | RMSE | MAPE (%) | MAE | RMSE | MAPE | MAE | RMSE | MAPE |
| VAR | 23.65 | 38.26 | 24.51 | 23.75 | 36.66 | 18.09 | 23.46 | 36.33 | 15.42 |
| FC-LSTM | 21.33 | 35.11 | 23.33 | 27.14 | 41.59 | 18.20 | 22.20 | 34.06 | 14.20 |
| DCRNN | 18.18 | 30.31 | 18.91 | 24.70 | 38.12 | 17.12 | 17.86 | 27.83 | 11.45 |
| STGCN | 17.49 | 30.12 | 17.15 | 22.70 | 35.55 | 14.59 | 18.02 | 27.82 | 11.40 |
| Graph WaveNet | 19.85 | 32.94 | 19.31 | 25.45 | 39.70 | 17.29 | 19.13 | 31.05 | 12.68 |
| ASTGCN | 17.69 | 29.66 | 19.40 | 22.93 | 35.22 | 16.56 | 18.61 | 28.16 | 13.08 |
| STSGCN | 17.48 | 29.21 | 16.78 | 21.19 | 33.65 | 13.90 | 17.13 | 26.80 | **10.96** |
| T-Graphormer | **15.88** | **24.58** | **15.62** | **20.40** | **30.37** | **13.86** | **16.88** | **24.61** | 11.43 |

Table 8: Horizon 12 forecasting results for the PEMS03, PEMS04, and PEMS08 datasets. We report the test metrics from the best configuration of T-Graphormer. The best-performing model for each metric is bolded.

Table 6. We confirm that on both datasets, models with added causal dilated convolution in their final prediction layers scale poorly. Specifically, the "causal mini" models consistently outperform the "causal small" models which outperform the "causal medium" model. This might be due to the added spatiotemporal bias in dilated causal convolution, which leads to easier overfitting as the model size grows. Another possible explanation is that since we fix the number of dilated convolution layers (3 layers with a dilation factor of 2) across model sizes, the larger embedding sizes in these models may require more dilated convolution layers to fully capture the context of the learned representations. In other words, the number of dilated convolution layers should grow with the embedding size.

Conversely, models with a simple linear final prediction layer scale better in the PEMS-BAY dataset where the "medium" models perform the second best, and the "small" models perform the third best. However, this comparison is not entirely fair since there are inconsistencies in our training configurations for medium-sized models. Specifically, we found that early-stopping and training the medium-sized models for 30 epochs instead of 50 works better on the PEMS-BAY dataset.

Finally, T-Graphormer prediction layer variants perform differently on the two datasets. The best-performing model on METR-LA utilizes causal dilated convolutions in its prediction layer. While this increases the number of parameters, we theorize that the additional temporal bias introduced by causal convolutions improves the model's prediction accuracy under limited training samples. However, it remains unclear why linear prediction layers perform poorly on the METR-LA dataset and why casual prediction layers perform poorly on the PEMS-BAY dataset.

The difficulty of predicting traffic conditions in the METR-LA dataset is also compounded by the presence of missing values. This is evident in Figures 3a and 3b. In Figure 3b, the traffic speed across all sensors abruptly drops to zero at approximately hours 8, 30, and 34. Such abrupt drops are absent in the PEMS-BAY dataset. Although we did not directly download the dataset, the simultaneous changes across all sensors in the METR-LA dataset strongly indicate missing values rather than traffic jams or road blockages.

To demonstrate the generalizability of our method, we evaluated it on three additional traffic prediction datasets introduced by Song et al. (2020) (see Table 7). The data preprocessing pipeline is similar to that used for PEMS-BAY and METR-LA, with the main difference being the data split ratio, which is set to 6 : 2 : 2 for these datasets. We report the prediction results for a horizon of 12 in Table 8. For all three datasets, we trained the smallest variant of T-Graphormer. The optimal hyperparameters for each dataset are detailed in Table 9.

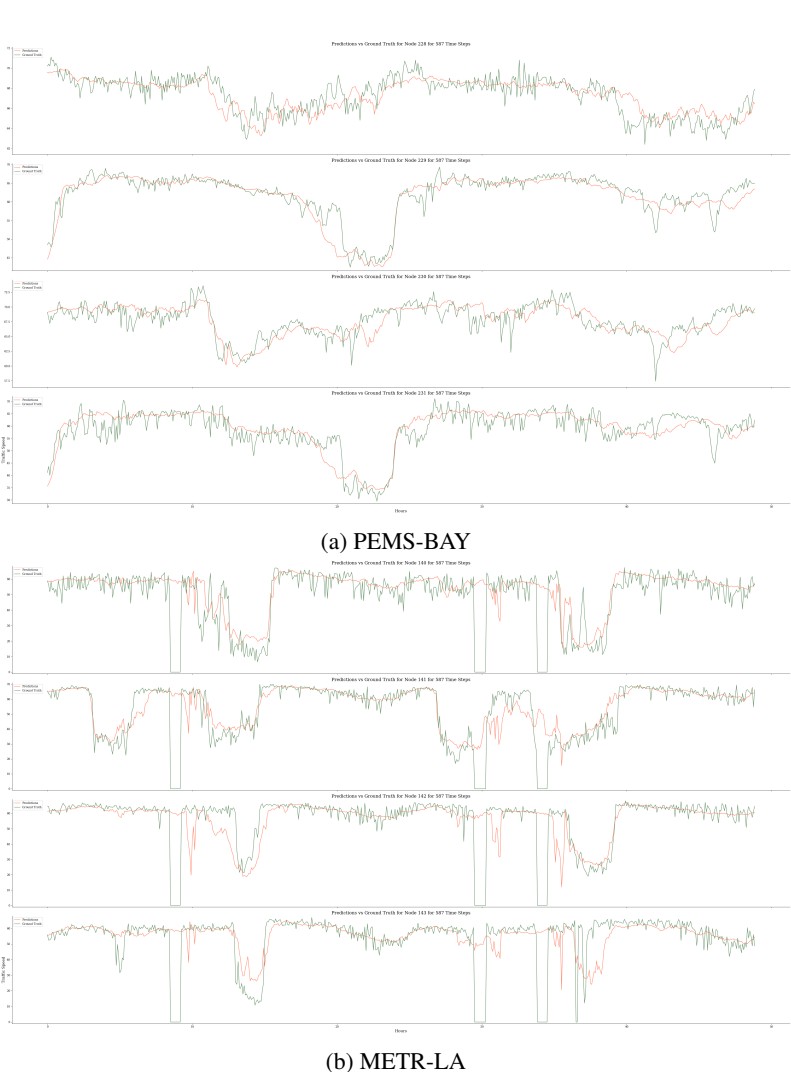

(a) PEMS-BAY

(b) METR-LA

Figure 3: Prediction visualizations of T-Graphormer on two datasets. (a) PEMS-BAY dataset. (b) METR-LA dataset. It is evident that T-Graphormer can learn the different traffic patterns between different traffic intersections in both datasets. For instance in subfigure (a), the traffic speed drop is at different time steps for all sensors. In subfigure (b), although all nodes have decreased traffic speeds at around hour 15 and hour 38, their rush hour behaviours are unique. It is also evident that T-Graphormer recognizes node 141 has an additional traffic speed drop at around hour 5. Note that in the METR-LA dataset, there are multiple time steps of missing values. It has become common practice to pad these missing values with the historical mean during data preprocessing, which is not done in this work.

| Configuration | Dataset | | |
|---|---|---|---|
| | PEMS03 | PEMS04 | PEMS08 |
| optimizer | AdamW (Loshchilov & Hutter, 2019) | | |
| optimizer momentum | $\beta_1, \beta_2 = 0.9, 0.999$ | | |
| learning rate schedule | cosine decay (Loshchilov & Hutter, 2017) | | |
| epochs | 100 | 100 | 100 |
| learning rate | 3.50e-3 | 7.50e-3 | 4.50e-3 |
| gradient clipping | 1.0 | 1.0 | 1.0 |
| weigh decay | 1e-5 | 1e-4 | 1e-6 |
| warmup epochs | 10 | 10 | 10 |
| batch size | 128 | 128 | 128 |
| dropout | 0.1 | 0.1 | 0.1 |
| layer-wise decay | 0.90 | 0.90 | 0.90 |
| # of parameters (M) | 1.77 | 1.69 | 1.48 |

Table 9: Best training hyperparameters of T-Graphormer mini on the additional datasets.

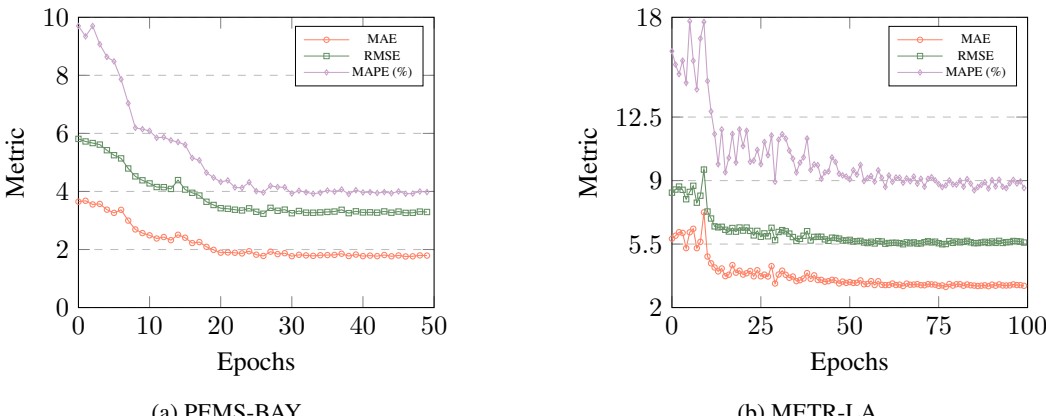

(a) PEMS-BAY

(b) METR-LA

Figure 4: Validation metrics on two traffic prediction datasets. The best-performing model is displayed.

On the PEMS03 dataset, T-Graphormer achieved improvements of **9.15%** in MAE, **15.85%** in RMSE, and **6.91%** in MAPE. On PEMS04, it improved MAE by **3.73%** and RMSE by **9.75%**. For PEMS08, T-Graphormer enhanced RMSE by **8.17%**. These results indicate a consistent trend: as the dataset size increases, T-Graphormer demonstrates stronger predictive performance. Notably, since the model is trained with the MSE loss, the most significant improvement is observed in the RMSE metric.

The baselines used for these datasets differ from those for PEMS-BAY and METR-LA due to variations in preprocessing and incomplete results reported in the original manuscripts of STEP, PDFormer, and STAEformer. Specifically, STEP reports results only for PEMS04, while PDFormer and STAEformer omit results for PEMS03, limiting the ability to validate their performance. We do not report results for the PEMS07 dataset because we are unable to fit the entire flattened graph sequence ($883 \times 12 = 10596$ tokens) into the memory of our GPUs during training.

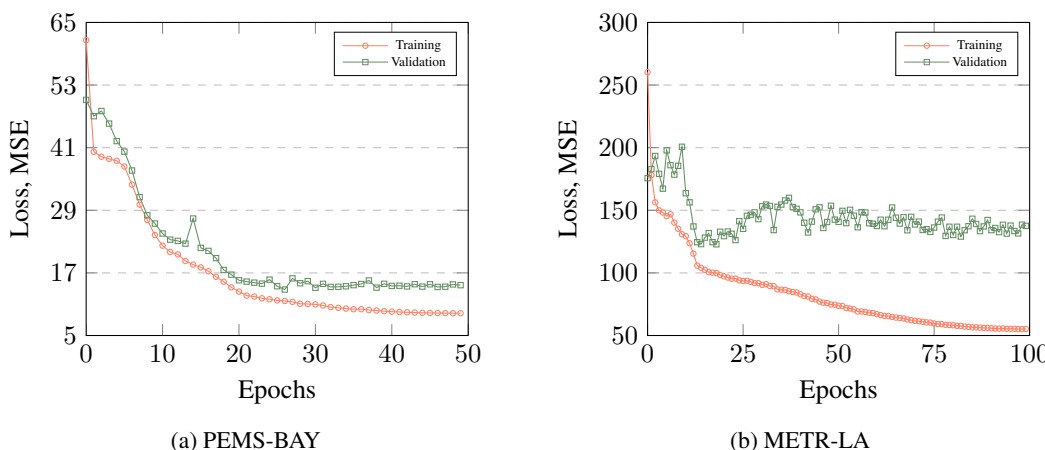

(a) PEMS-BAY

(b) METR-LA

Figure 5: Training and validation loss of the best-performing models on two traffic prediction datasets. For model Training loss is plotted as the global average per epoch, and validation loss is computed as the average per epoch. Both losses are mean squared errors. For PEMS-BAY, the gap between training and validation is much closer than that of METR-LA.

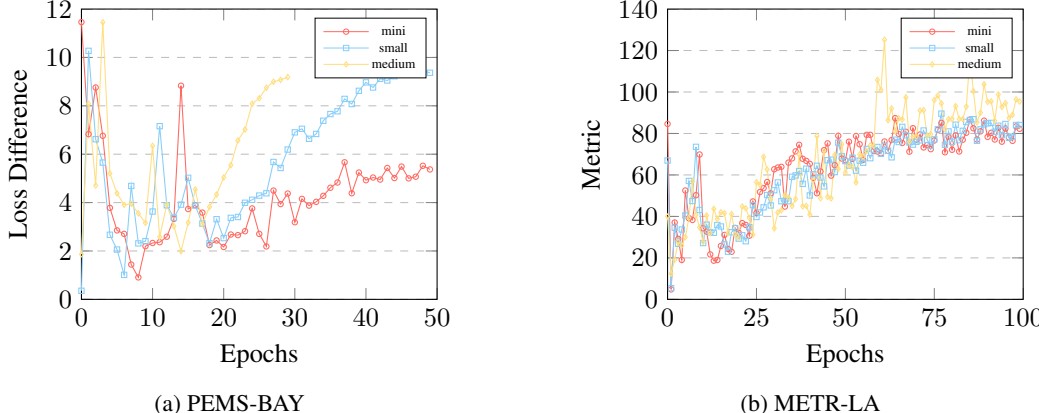

(a) PEMS-BAY

(b) METR-LA

Figure 6: Mean squared error differences of the best-performing models on the respective training and validation dataset. Compared to the mini and small models, the medium model on PEMS-BAY only has 30 epochs since it is the best configuration we found.

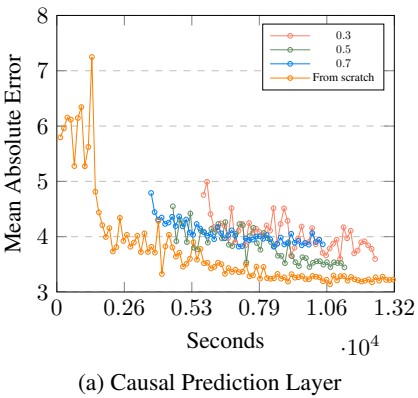
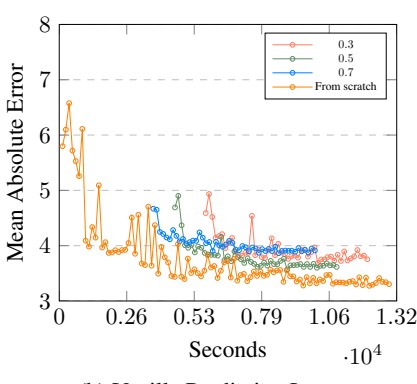

(a) Causal Prediction Layer        (b) Vanilla Prediction Layer

Figure 7: Performance comparison on the METR-LA traffic prediction dataset between masked self-supervised pre-training and training from scratch. Three masking ratios are evaluated for each prediction layer architecture. Each data point represents one epoch on the validation set. For pre-trained methods, the wall-clock time is calculated by adding the total pre-training time to the per-epoch training time (129 seconds for the vanilla prediction layer and 137 seconds for the causal prediction layer). Notably, higher masking ratios result in reduced pre-training time due to fewer tokens per batch. Since the METR-LA dataset is quite small, further experiments on larger datasets are needed to demonstrate the full pre-training potential.

