# OpenReview forum: "T-Graphormer: Using Transformers for Spatiotemporal Forecasting"
_ICLR.cc/2025/Conference — Submitted to ICLR 2025_

### Official Review · Reviewer_vhLu · 2024-10-26

**Soundness:** 2
**Presentation:** 4
**Contribution:** 2
**Rating:** 5
**Confidence:** 4

**Summary:**

The paper presents the Temporal-Graphormer (T-Graphormer) framework, which encodes spatial-temporal data in the temporal dimension and employs the attention mechanism to effectively capture complex spatial and temporal relationships. Comprehensive experiments also demonstrate its effectiveness.

**Strengths:**

1. The time series forecasting problem is a valuable and impactful area of research, with significant applications across various fields.
2. This paper introduces a centrality encoding and positional encoding combination approach, with the attention mechanism, to capture complex spatial-temporal correlations. This approach can be easily adapted to various domains.
3. This paper provides a clear presentation and thorough description of the methodology and experimental setup.

**Weaknesses:**

1. This paper claims that T-Graphormer performs well on spatial-temporal data; however, only traffic datasets were used in the experiments, so testing on additional datasets is encouraged. Additionally, the experiments focus primarily on main results and ablation studies, and further experiments, such as hyperparameter analysis, are recommended.
2. This paper would benefit from incorporating more recent citations on state-of-the-art Transformer research, particularly in the baseline comparisons. Adding newer relevant works, such as DSformer [Yu, 2023] and LDT [Feng, 2024], is recommended.
3. Even though detailed information about the implementation are provided, no code provided would still make it hard to evaluate the results.

Reference:
[1] Feng, S., Miao, C., Zhang, Z., & Zhao, P. (2024). Latent Diffusion Transformer for Probabilistic Time Series Forecasting. Proceedings of the AAAI Conference on Artificial Intelligence, 38(11), 11979-11987. https://doi.org/10.1609/aaai.v38i11.29085.
[2] Yu, Chengqing, et al. "Dsformer: A double sampling transformer for multivariate time series long-term prediction." Proceedings of the 32nd ACM international conference on information and knowledge management. 2023.

**Questions:**

1. The encoding strategy in the methodology seems primarily focused on a single data dimension. However, spatial-temporal data inherently involves complex interactions between spatial and temporal dimensions. Given that the embedding preprocessing is focused solely on the temporal dimension, how could the current method effectively  capture dependencies across both dimensions?
2. The attention mechanism is indeed effective at capturing correlations, but have you considered testing the encoding strategy with alternative mechanisms, such as replacing attention with a Graph Neural Network (GNN), in ablation studies? This would help address the lack of sufficient Transformer or attention-based baselines for comparison in the experiments.
3. Could you provide a brief summary of how the graph was generated and the adjacency matrix constructed? While I understand the implementation follows another paper, it would be helpful if these details were mentioned directly in your paper for clarity.

---

> ### Author Response · Authors · 2024-11-22
> **Author Responses**
>
> Thank you for dedicating your time to carefully review our work and provide valuable feedback! We are glad you also believe in using this approach in other domains. Here, we address your comments in detail:
> >This paper claims that T-Graphormer performs well on spatial-temporal data; however, only traffic datasets were used in the experiments, so testing on additional datasets is encouraged. Additionally, the experiments focus primarily on main results and ablation studies, and further experiments, such as hyperparameter analysis, are recommended.
>
> * **Dataset Selection**: We acknowledge the importance of evaluating T-Graphormer across a broader set of traffic datasets. We are actively working to include PEMS03, PEMS04, PEMS07, and PEMS08 in our experiments. The results will be updated in the revised manuscript once these evaluations are complete.
> * **Hyperparameter Study** We have conducted extensive hyperparameter testing. For instance, in Tables 4 and 5 of the Appendix, we report the best hyperparameters of different T-Graphromer model variants on the two traffic datasets. In Table 6, we report the test metrics of these variants. We also find that larger variants of T-Graphormer are likely to overfit, and this behaviour is consistent with the recent findings on the empirical scaling laws for training LLMs [1].
>
> >This paper would benefit from incorporating more recent citations on state-of-the-art Transformer research, particularly in the baseline comparisons. Adding newer relevant works, such as DSformer [Yu, 2023] and LDT [Feng, 2024], is recommended.
>
> We apologize for the oversight in not including these recent related works in our literature review and baseline comparisons. We have incorporated advanced baselines, including PDFormer and STAEformer, into our comparisons (see Figure 2 and Section 2.1). The results demonstrate that T-Graphormer consistently outperforms these models.
>
> >Even though detailed information about the implementation are provided, no code provided would still make it hard to evaluate the results.
>
> As stated in our Reproducibility Statement, we include the ave included links to the anonymized code repository, the datasets, and
> the pre-trained model weights in Section A.1.
>
> >Q1. The encoding strategy in the methodology seems primarily focused on a single data dimension. However, spatial-temporal data inherently involves complex interactions between spatial and temporal dimensions. Given that the embedding preprocessing is focused solely on the temporal dimension, how could the current method effectively capture dependencies across both dimensions?
>
> In fact, T-Graphormer has more structural encoding methods in the spatial dimension.
> * By incorporating the centrality encoding into node features, and using the shortest path distance to adjust attention bias, the model can update representation by selectively paying attention to more important nodes.
> * In the temporal dimension, we add learnable positional encoding to the initial node features. This has proven useful in the original Transformer architecture [2], and the ViT architecture [3, 4] where data structures are more complex.
>
> >Q2. The attention mechanism is indeed effective at capturing correlation ... This would help address the lack of sufficient Transformer or attention-based baselines for comparison in the experiments.
>
> This has been done in a similar manner by STSGCN [5], where they also try to simultaneously capture spatial and temporal dependencies with GNNs. They do so by connecting the nodes across time steps. In comparison, T-Graphormer processes the input as a sequence. This distinction underscores a fundamental difference in how the two models integrate spatial and temporal information.
>
> >Could you provide a brief summary of how the graph was generated and the adjacency matrix constructed? While I understand the implementation follows another paper, it would be helpful if these details were mentioned directly in your paper for clarity.
>
> To construct the sensor graph, we compute the pairwise road network distances between sensors and build the adjacency matrix using thresholded Gaussian kernel. We have added these details in our appendix.
>
> We sincerely thank the reviewer for the thoughtful and detailed feedback, which has significantly improved our manuscript in terms of its clarity, rigour, and impact. We hope our detailed responses and manuscript revisions address all concerns. We look forward to further correspondence.
>
> [1]Kaplan, J. (2020). Scaling laws for neural language models. arXiv preprint.
>
> [2]Vaswani, A. (2017). Attention Is All You Need. NeurIPS.
>
> [3]Dosovitskiy, A. (2020). An image is worth 16x16 words: Transformers for image recognition at scale. ICLR.
>
> [4]Feichtenhofer, C. (2022). Masked autoencoders as spatiotemporal learners. NeurIPS.
>
> [5]Song, C. (2020). Spatial-temporal synchronous graph convolutional networks: A new framework for spatial-temporal network data forecasting. AAAI.

---

> ### Comment · Reviewer_vhLu · 2024-11-25
>
> Thanks the authors for their efforts. I have carefully read all the replies and decided to maintain my score.  I acknowledge your efforts to incorporate additional traffic datasets (PEMS03, PEMS04, etc.). Including these datasets will indeed enhance the generalizability and robustness of the proposed method. However, as these results are not yet included, the current evaluation remains limited to 2 traffic datasets, which constrains the paper's broader applicability.

---

> > ### Author Response · Authors · 2024-11-27
> >
> > Dear Reviewer vhLu,
> >
> > Thank you for taking the time to carefully read all the replies.
> >
> > We have also included the results of these datasets in our updated manuscript. Please let us know if you have any additional concerns or questions.
> >
> >
> > Best,
> > Authors

---

> > ### Author Response · Authors · 2024-12-02
> >
> > Dear Reviewer vhLu,
> >
> >
> > As the review period draws to a close, we wanted to kindly remind you that we have addressed your valuable comments in our recent responses. We greatly appreciate your insights and would be happy to discuss further if you feel there are any remaining concerns or areas that need clarification.
> >
> > Looking forward to hearing from you!
> >
> > Best regards,
> >
> > Authors

---

### Official Review · Reviewer_nhKS · 2024-10-28

**Soundness:** 2
**Presentation:** 2
**Contribution:** 1
**Rating:** 5
**Confidence:** 4

**Summary:**

In this paper, the authors develop a spatial temporal forecasting/traffic prediction task with transformer-based structure. Unlike other methods, T-Graphformer can capture spatial-temporal correlations directly and it has a better performance than some of the baseline methods.

**Strengths:**

1. Well written related work.
2. Nice figure 2. It clearly explains the model framework.
3. The ablation study is somehow strong and it shows that different model components are useful.

**Weaknesses:**

1. The baseline methods are outdated. Authors need to compared this model with more advanced and latest baselines, instead of SVM and FC-LSTM. Additionally, more datasets should be included. For example, you might include STID, MegaCRN and PDFormer.
References:
Zezhi Shao, Zhao Zhang, Fei Wang, Wei Wei, and Yongjun Xu. 2022. Spatialtemporal identity: A simple yet effective baseline for multivariate time series forecasting. In Proceedings of the 31st ACM International Conference on Information
& Knowledge Management. 4454–4458.
Renhe Jiang, Zhaonan Wang, Jiawei Yong, Puneet Jeph, Quanjun Chen, Yasumasa Kobayashi, Xuan Song, Shintaro Fukushima, and Toyotaro Suzumura. 2023. Spatio-temporal meta-graph learning for traffic forecasting. In Proceedings of the AAAI Conference on Artificial Intelligence, Vol. 37. 8078–8086.
Jiawei Jiang, Chengkai Han, Wayne Xin Zhao, and Jingyuan Wang. 2023. PDFormer: Propagation Delay-aware Dynamic Long-range Transformer for Traffic Flow Prediction. In AAAI. AAAI Press.
For datasets, more traffic datasets, such as PEMS03, PEMS04, PEMS07 and PEMS08 can be included. It is not sufficient to only use two datasets to demonstrate the effectiveness of your model.
2. The writing should be improved. The authors use 'spatial temporal forecasting', 'multivariate time series forecasting', 'traffic prediction' in the introduction and section. Although these terms are correlated, they are different and they might be better to focus on one specific task while writing this paper. Since your data is traffic, it is better to frame it as a traffic prediction problem and emphasize it in your introduction.
3. The methodology itself is not innovative. For example, authors do not make modifications to the traditional transformer structure to make improvement. Many papers adjust the attention mechanism to better capture temporal/spatial correlations. Why you only use the original transformer?

**Questions:**

1. See weaknesses.
2. Why you choose STEP as the baseline? It seems that your model does not have a higher performance.
3. Since your title is spatial temporal forecasting, do you agree that more spatial temporal dataset should be concluded, such as weather dataset and crime dataset?

---

> ### Author Response · Authors · 2024-11-22
> **Author Responses**
>
> Thank you for dedicating your time to carefully review our work and provide valuable feedback! We are glad you like Figure 2 and the ablation study. Here, we address your comments in detail:
>
>
> >The baseline methods are outdated. Authors need to compared this model with more advanced and latest baselines, instead of SVM and FC-LSTM. Additionally, more datasets should be included. For example, you might include STID, MegaCRN and PDFormer ... For datasets, more traffic datasets, such as PEMS03, PEMS04, PEMS07 and PEMS08 can be included. It is not sufficient to only use two datasets to demonstrate the effectiveness of your model.
>
> * **Baseline Comparisons**: We apologize for the oversight in not including these recent related works in our literature review and baseline comparisons. We have incorporated advanced baselines, including PDFormer and STAEformer, into our comparisons (see Figure 2 and Section 2.1). The results demonstrate that T-Graphormer consistently outperforms these models, highlighting the robustness and effectiveness of our proposed method.
> * **Dataset Selection**: We acknowledge the importance of evaluating T-Graphormer across a broader set of traffic datasets. We are actively working to include PEMS03, PEMS04, PEMS07, and PEMS08 in our experiments. The results will be updated in the revised manuscript once these evaluations are complete.
>
> >The writing should be improved. The authors use 'spatial temporal forecasting', 'multivariate time series forecasting', 'traffic prediction' in the introduction and section. Although these terms are correlated, they are different and they might be better to focus on one specific task while writing this paper. Since your data is traffic, it is better to frame it as a traffic prediction problem and emphasize it in your introduction.
>
> We would appreciate clarification on which section the reviewer finds unclear.
>
> We have carefully written our introduction such that the scope of discussion is gradually narrowed to traffic prediction, which is the specific task we are evaluating T-Graphormer on. We believe that since T-Graphormer is a Transformer architecture directly applied onto traffic data with minimal spacetime inductive biases, it can be readily applied to other data modalities. In our work, we are introducing this general framework. This has been made clearer in Section 6 of our manuscript.
>
> > The methodology itself is not innovative. For example, authors do not make modifications to the traditional transformer structure to make improvement. Many papers adjust the attention mechanism to better capture temporal/spatial correlations. Why you only use the original transformer?
>
> We respectfully disagree with the assertion that the methodology lacks innovation. We not only borrow the structural encoding methods from Graphormer and extend them along the temporal dimension but also examine the effects of added special tokens (namely $\textt{cls} and \textt{graph}) and prediction layers (fully connected and dilated causal). Specifically, the attention mechanism is adjusted in our spatial encoding method (defined in Equation 7). We decide to use the original Transformer with added structural encoding methods to fully utilize its global attention mechanism such that the spatial and temporal inter-correlations are modelled simultaneously.
>
> >Why you choose STEP as the baseline? It seems that your model does not have a higher performance.
>
> We selected STEP as a baseline because it has been shown to outperform PDFormer and STAEformer on the PEMS-BAY and METR-LA datasets. Our results demonstrate that T-Graphormer consistently outperforms STEP in predicting the next 12 time steps (1-hour window), which we believe underscores the model's efficacy for longer-range traffic predictions. We discuss the performance gap in short-range predictions in Section 5.3 of our manuscript.
>
> >Since your title is spatial temporal forecasting, do you agree that more spatial temporal dataset should be concluded, such as weather dataset and crime dataset?
>
> We agree that T-Graphormer should be evaluated on other spatiotemporal datasets, such as weather and crime datasets. While our current scope focuses on introducing this framework for traffic prediction, future work will explore its applicability to diverse spatiotemporal data modalities.
>
>
> We sincerely thank the reviewer for the thoughtful feedback, which has significantly improved our manuscript in terms of its clarity, rigour, and impact. We hope our detailed responses and manuscript revisions address all concerns. We look forward to further correspondence.

---

> > ### Comment · Reviewer_nhKS · 2024-11-24
> >
> > Thanks for your prompt response and I decide to raise my score to 5.

---

> > > ### Author Response · Authors · 2024-11-26
> > >
> > > Dear Reviewer nhKS,
> > >
> > > Thank you for your prompt response as well and for deciding to raise your score to 5. We appreciate your thoughtful evaluation and the time you’ve dedicated to reviewing our work.
> > >
> > > To continue improving our work, we would be grateful if you could share any further feedback or suggestions. Are there additional areas where we could focus our efforts?
> > >
> > > Thank you again for your time and support.

---

> > > ### Author Response · Authors · 2024-12-02
> > >
> > > Dear Reviewer nhKS,
> > >
> > > As the review period draws to a close, we wanted to kindly remind you that we have addressed your valuable comments in our recent responses. We greatly appreciate your insights and would be happy to discuss further if you feel there are any remaining concerns or areas that need clarification.
> > >
> > >
> > > Looking forward to hearing from you!
> > >
> > > Best regards,
> > >
> > > Authors

---

> > > > ### Comment · Reviewer_nhKS · 2024-12-02
> > > >
> > > > Thanks for your responses. I will discuss with other reviewers in the next stage.

---

### Official Review · Reviewer_JiSB · 2024-11-03

**Soundness:** 1
**Presentation:** 2
**Contribution:** 1
**Rating:** 3
**Confidence:** 5

**Summary:**

In this paper, the authors propose an approach called Temporal Graphormer(T-Graphormer). It can models spatiotemporal correlations directly by capturing spatiotemporal correlations and adding special tokens.

**Strengths:**

S1. The author has effectively enhanced Graphormer for spatiotemporal prediction, incorporating improvements in both embeddings and tokens.

S2. T-Graphormer demonstrates superior performance, particularly when forecasting the next 12 time steps (1-hour window).

**Weaknesses:**

Weaknesses:

W1.Key points lack details. While the existing content and formulas in Graphormer are well-articulated, The description of the novel contributions is not sufficiently detailed. In Section 4.2, the paper only presents the modified transformer method for T-Graphormer only in brief with the expression l=T'*N. In Section 4.3, the paper doesn't clearly convey the formulas for special tokens. Could you provide detailed mathematical formulas and subsequent explanations for these original contributions?

W2. Lack of recent related work. The related work mentioned in Chapter 2, Related Works, basically references studies prior to 2020. There's also a lack of references to newer traffic prediction models. Including recent works such as  STAEformer [1]  would strengthen the baselines.

[1]Liu H, Dong Z, Jiang R, et al. Spatio-temporal adaptive embedding makes vanilla transformer sota for traffic forecasting[C]//Proceedings of the 32nd ACM international conference on information and knowledge management. 2023: 4125-4129.

W3. Lack of visualization. While the main claim of the paper is that directly modeling spatiotemporal correlations is better than modeling them separately, there is no accompanying visualization or case study to substantiate this claim, reducing the argument's credibility.

Could you provide a visualization of an attention example for a particular sensor to illustrate that the T-Graphormer indeed learns spatiotemporal patterns effectively compared to previous methods?

Minor remarks:

* In the last paragraph of 4.2, the symbol upright phi appears without prior definition.

**Questions:**

See Weaknesses.

---

> ### Author Response · Authors · 2024-11-22
> **Author Responses**
>
> Thank you for dedicating your time to carefully review our work and provide valuable feedback! Here, we address your comments in detail:
>
> >W1.Key points lack details. While the existing content and formulas in Graphormer are well-articulated, The description of the novel contributions is not sufficiently detailed. In Section 4.2, the paper only presents the modified transformer method for T-Graphormer only in brief with the expression l=T'*N. In Section 4.3, the paper doesn't clearly convey the formulas for special tokens. Could you provide detailed mathematical formulas and subsequent explanations for these original contributions?
>
> * In Section 4.2, we not only present the flattening process of the graph sequence ($l = T' \times N$), but also explicitly detail the structural and positional encoding extensions. We accompany these descriptions with Figure 2, visualizing the model architecture with carefully selected colours matching that of nodes in Figure 1. We also highlight the novelty of these encodings compared to the original Graphormer formulation.
> * We apologize for not elaborating on the details of the special tokens and causing confusion. To clarify (we have added these details in our revised manuscript):
>     * The $\texttt{cls}$ token is implemented similarly to the virtual node in Graphormer. Its purpose is to aggregate information across the entire sequence and propagate it back to individual tokens.
>     * The $\texttt{graph}$ token functions analogously to the $\texttt{sep}$ token in BERT (Devlin et al., 2018) [2], serving as a delimiter for graph signals at different time steps.
>
> >W2. Lack of recent related work. The related work mentioned in Chapter 2, Related Works, basically references studies prior to 2020. There's also a lack of references to newer traffic prediction models. Including recent works such as STAEformer [1] would strengthen the baselines.
>
> We apologize for the oversight in not including these recent related works in our literature review and baseline comparisons. We have added PDFormer and STAEformer in Figure 2 and Section 2. 1. These additions position our work within the broader context of modern traffic prediction models and highlight the improvements our method achieves over these benchmarks.
>
> >W3. Lack of visualization. While the main claim of the paper is that directly modeling spatiotemporal correlations is better than modeling them separately, there is no accompanying visualization or case study to substantiate this claim, reducing the argument's credibility. Could you provide a visualization of an attention example for a particular sensor to illustrate that the T-Graphormer indeed learns spatiotemporal patterns effectively compared to previous methods?
>
> We definitely agree that visualizing the model's prediction can provide further credibility and a more intuitive understanding of its performance. Since we flatten the entire graph sequence, we find it extremely difficult to interpret the attention scores on 3900 ($325 \times 12$) tokens in PEMS-BAY and 2484 ($207 \times 12$) tokens in METR-LA. We instead provide a more interpretable visualization in Figures 3 and 4. These figures demonstrate T-Graphormer's spatiotemporal modelling capability by showcasing predictions for individual sensors across 587 time steps. This alternative approach offers a practical perspective on the model's performance.
>
> >Minor remarks: In the last paragraph of 4.2, the symbol upright phi appears without prior definition.
>
> Thank you for spotting our mistake, this has been corrected in our manuscript.
>
> We sincerely thank the reviewer for the thoughtful feedback, which has significantly improved our manuscript in terms of its clarity, rigour, and impact. We hope our detailed responses and manuscript revisions address all concerns. We look forward to further correspondence.

---

> > ### Comment · Reviewer_JiSB · 2024-11-24
> > **Response from  JiSB**
> >
> > Thanks for your efforts. I have carefully read all the rebutalls and decided to maintain my score as it is (below the acceptance bar).

---

> > > ### Author Response · Authors · 2024-11-26
> > >
> > > Dear Reviewer JiSB,
> > >
> > > Thank you for your efforts in reviewing our work and carefully reading our rebuttals. While we respect your decision to maintain your score, we would greatly appreciate any additional feedback you could provide to help us better understand your concerns.
> > >
> > > Were there specific points in our rebuttals that you felt did not fully address your initial concerns? Additionally, could you elaborate on the key reasons for maintaining your current score? Understanding this would be invaluable for our work as we strive to improve it and address any remaining gaps.
> > >
> > > Thank you again for your time and thoughtful review.

---

> > > ### Author Response · Authors · 2024-12-02
> > >
> > > Dear Reviewer JiSB,
> > >
> > > As the review period draws to a close, we wanted to kindly remind you that we have addressed your valuable comments in our recent responses. We greatly appreciate your insights and would be happy to discuss further if you feel there are any remaining concerns or areas that need clarification.
> > >
> > >
> > > Looking forward to hearing from you!
> > >
> > > Best regards,
> > >
> > > Authors

---

### Official Review · Reviewer_SDZa · 2024-11-04

**Soundness:** 2
**Presentation:** 1
**Contribution:** 1
**Rating:** 1
**Confidence:** 4

**Summary:**

The manuscript focuses on spatiotemporal forecasting. To simultaneously learn both spatial and temporal correlations, the authors propose T-Graphormer based on the transformer architecture and design temporal and spatial encoding methods for this purpose.

**Strengths:**

Two novel spatial feature extraction methods are proposed in this paper, one for encoding the in-degree and out-degree of the graph structure and the other for encoding the shortest travel distance between graph nodes.

**Weaknesses:**

1. There have been several methods [1][2] that apply transformer architecture to spatio-temporal forecasting. The idea of using transformers to simultaneously capture temporal and spatial correlations is not sufficiently novel.
2. Section 4.3 does not adequately explain the setting of additional tokens. These tokens are typically used in the NLP domain, but the authors fail to clarify their significance in spatio-temporal forecasting or provide details on their implementation.
3. The experimental design has shortcomings. First, the dataset selection is insufficient. The study focuses on spatio-temporal forecasting but only uses two traffic speed datasets to validate the model, which is inadequate to demonstrate the method’s superiority. Second, the experimental baseline lacks comparisons with advanced transformer-based methods [1][2]. Finally, the hyperparameter experiments do not sufficiently explore hyperparameter values.
4. The manuscript's structure is also not well-balanced. Chapter 4, which describes the main method, occupies only one and a half pages, while other sections take up considerably more space.
5. The language presentation is rough. The adjacency matrix dimension in Section 3.1 is incorrect. Section 4.1 consists of only two sentences and lacks a period. Additionally, the summary in Section 6 is confusing and poorly presented.

Reference:
[1] Liu H, Dong Z, Jiang R, et al. Spatio-temporal adaptive embedding makes vanilla transformer sota for traffic forecasting[C]//Proceedings of the 32nd ACM international conference on information and knowledge management. 2023: 4125-4129.
[2] Jiang J, Han C, Zhao W X, et al. Pdformer: Propagation delay-aware dynamic long-range transformer for traffic flow prediction[C]//Proceedings of the AAAI conference on artificial intelligence. 2023, 37(4): 4365-4373.

**Questions:**

Please refer to the weaknesses.

---

> ### Author Response · Authors · 2024-11-22
> **Author Responses**
>
> Thank you for dedicating your time to review our work and provide valuable feedback. We address your comments in detail:
>
> >Strengths: Two novel spatial feature extraction methods are proposed in this paper, one for encoding the in-degree and out-degree of the graph structure and the other for encoding the shortest travel distance between graph nodes.
>
> While these structural encoding techniques are crucial to T-Graphormer's modelling capabilities when the graph sequence is flattened (see Section 5.4, Ablation Studies), we did not introduce these methods. They were first proposed in Graphormer by Ying et al. (2021) [1]. As stated in Section 4, our contribution extends these structural encoding methods to the **temporal dimension**, building on their foundation.
>
> >There have been several methods [1][2] that apply transformer architecture to spatio-temporal forecasting. The idea of using transformers to simultaneously capture temporal and spatial correlations is not sufficiently novel.
>
> We respectfully disagree. While prior methods leverage the Transformer architecture in their encoder models, T-Graphormer is **fundamentally different in how it unifies spatiotemporal learning**.
>
> As outlined in Section 2.1 (Related Work), existing methods use separate modules for temporal and spatial learning. Regarding the references the reviewer provided,
> * PDFormer splits attention heads into *Semantic Spatial Attention*, *Geographic Spatial Attention*, and *Temporal Self-Attention*.
> * STAEformer uses *Temporal transformer layer* followed by a *Spatial transformer layer* (Equation 7).
>
> Our work directly applies the Transformer architecture to spatiotemporal data, acknowledging that dependencies exist across dimensions, not just within them. By treating spatiotemporal data holistically, T-Graphormer effectively models these interdependent correlations without separating temporal and spatial dimensions.
>
> >Section 4.3 does not adequately explain the setting of additional tokens. These tokens are typically used in the NLP domain, but the authors fail to clarify their significance in spatio-temporal forecasting or provide details on their implementation.
>
> We acknowledge the need for clearer explanations of the additional tokens.
> * The $\texttt{cls}$ token is implemented similarly to the virtual node in Graphormer. Its purpose is to aggregate information across the entire sequence and propagate it back to individual tokens.
> * The $\texttt{graph}$ token functions analogously to the $\texttt{sep}$ token in BERT (Devlin et al., 2018) [2], serving as a delimiter for graph signals at different time steps.
>
> >The experimental design has shortcomings. First, the dataset selection is insufficient ... Finally, the hyperparameter experiments do not sufficiently explore hyperparameter values.
>
> 1. **Dataset Selection**: We are actively expanding our experiments to include additional traffic speed datasets and will update our results accordingly.
> 2. **Baseline Comparisons**: Advanced Transformer-based baselines have been added to Table 2, and T-Graphormer maintains superior performance on the PEMS-BAY dataset and METR-LA dataset. We apologize for the oversight in not including these recent related works in our literature review and baseline comparisons.
> 3. **Hyperparameter Exploration**: We are unsure which specific hyperparameter values the reviewer is referring to. Additional clarification would help us refine this aspect.
>
> >The manuscript's structure is also not well-balanced. Chapter 4, which describes the main method, occupies only one and a half pages, while other sections take up considerably more space.
>
> We believe there is no universally "correct" manuscript structure. Our framework is designed to be simple and intuitive, warranting a concise description in Chapter 4. The remaining sections focus on contextualizing, validating, and analyzing the method. However, if there are specific aspects of Chapter 4 that the reviewer finds unclear, we are happy to provide further elaboration.
>
> >The language presentation is rough. The adjacency matrix dimension in Section 3.1 is incorrect. Section 4.1 consists of only two sentences and lacks a period. Additionally, the summary in Section 6 is confusing and poorly presented.
>
> We sincerely apologize for these errors and have corrected them in the updated manuscript. Regarding Section 6, we would greatly appreciate specific feedback on which parts are confusing to address them more effectively. Nonetheless, we have revised and improved the clarity of this section.
>
> Once again, we appreciate your feedback and look forward to further correspondence.
>
> [1]. [Do transformers really perform badly for graph representation?](https://proceedings.neurips.cc/paper/2021/file/f1c1592588411002af340cbaedd6fc33-Paper.pdf). (Ying et al., NeurIPS 2021)
>
> [2]. [BERT: Pre-training of Deep Bidirectional Transformers for Language Understanding](https://aclanthology.org/N19-1423) (Devlin et al., NAACL 2019)

---

> > ### Comment · Reviewer_SDZa · 2024-11-27
> >
> > Thank you for your detailed reply. Regarding the hyperparameter experiment, I would like to point out that only two values are set for each hyperparameter. This limited search space may not be sufficient to identify the optimal parameters and also restricts a comprehensive evaluation of the impact of hyperparameters on the model's performance.
> >
> > While I acknowledge the addition of appropriate experiments to validate the method's effectiveness and the improvements made in the presentation, the overall contribution of this work remains limited and the performance gains are relatively modest. Therefore, I decided to increase the score to 3.

---

> ### Author Response · Authors · 2024-11-26
>
> Dear Reviewer SDZa,
>
> As the discussion period comes to an end, we would love to have additional feedback on our rebuttal response so we can continue improving the manuscript.
>
> Have our comments addressed your concerns about our work? Do you have further questions?
>
> Thank you again for your time.
>
> Best,
> Authors

---

> ### Author Response · Authors · 2024-11-27
>
> Dear Reviewer SDZa,
>
> Thank you for your response. We sincerely appreciate your feedback and the opportunity to further clarify our contributions.
>
>
> * **Hyperparameter Experiment**: We apologize if our description led to any misunderstanding. As detailed in Tables 4 and 5, for each dataset and model variant (mini, small, medium, causal mini, causal small, causal medium), we conducted an extensive hyperparameter search. Specifically, we evaluated the following parameters:
>
>    *learning rate, gradient clipping, weight decay, dropout, and layerwise decay*
>
>     Among these, we observed that learning rate, layerwise decay, and weight decay (in that order) had the most significant impact on model performance during training.
>
>     If you feel that further details are necessary, we would be happy to expand on this as we stored all of our runs on Weights and Biases.
>
> * **Contribution**:  To the best of our knowledge, this work is the first to directly apply a Transformer architecture to spatiotemporal data with minimal spacetime inductive biases.
>
>     Our proposed method, T-Graphormer, not only improves RMSE by 10% on the PEMS-BAY dataset and MAPE by 10% on METR-LA compared to existing approaches but also reframes spatiotemporal forecasting. By modelling spatiotemporal data holistically, T-Graphormer identifies interdependent correlations without explicitly separating temporal and spatial dimensions. Future research can easily extend T-Graphormer by incorporating domain knowledge into the architecture, such as using adaptive spatiotemporal embeddings from STAEformer.
>
> We thank you again for your feedback, and we hope this response addresses your concerns. Thank you again for your time.
>
> Best,
>
> Authors

---

> > ### Author Response · Authors · 2024-12-02
> >
> > Dear Reviewer SDZa,
> >
> > As the review period draws to a close, we wanted to kindly remind you that we have addressed your valuable comments in our recent responses. We greatly appreciate your insights and would be happy to discuss further if you feel there are any remaining concerns or areas that need clarification.
> >
> >
> > Additionally, we noticed that the score has not been updated, and we would like to ensure that all your questions have been resolved before finalizing your review.
> >
> >
> > Looking forward to hearing from you!
> >
> > Best regards,
> >
> > Authors

---

### Author Response · Authors · 2024-11-22
**Phase 1 Comment to Address Some Common Questions from All Reviewers**

We sincerely thank all reviewers for their valuable feedback and thoughtful comments, which have significantly helped us improve our work. Below, we address the common concerns raised across the reviews, including baseline comparisons, dataset selection, and the novelty of our method.

**Baseline Comparisons**

Despite STEP (Shao et al., 2022) having a Transformer-based architecture, we acknowledge the need for more comprehensive comparisons with more recent Transformer-based baselines. In our revised manuscript, we have incorporated advanced models such as PDFormer (Jiang et al., 2023) and STAEformer (Liu et al., 2023) into our experiments. These additions, along with detailed results in Tables 2 and 3, demonstrate that T-Graphormer still consistently achieves state-of-the-art performance on traffic datasets such as METR-LA and PEMS-BAY. These updates strengthen our claims about the effectiveness of T-Graphormer and its ability to model spatiotemporal data holistically.

**Dataset Selection**


We understand the importance of evaluating T-Graphormer across a diverse set of datasets. In our initial submission, we focused on METR-LA and PEMS-BAY due to their prevalence in traffic prediction studies and their challenging properties. However, in response to reviewers' feedback, we are actively expanding our experiments to include additional traffic datasets such as PEMS03, PEMS04, PEMS07, and PEMS08. Furthermore, we are exploring the potential of applying T-Graphormer to other spatiotemporal domains, such as weather and crime datasets, and will include these results in future work.

**Novelty**

We clarify that T-Graphormer introduces significant innovations in spatiotemporal modelling:

* *Unified Spatiotemporal Learning*: Unlike existing Transformer-based methods (e.g., STAEformer, PDFormer) that separate spatial and temporal learning into distinct modules, T-Graphormer treats spatiotemporal data as a flattened sequence. This approach leverages the global attention mechanism of Transformers to simultaneously model interdependencies across both dimensions.
* *Extended Structural Encoding*: While we adopt the centrality and shortest-path-distance encodings from Graphormer, our novel contribution lies in extending these encodings to the temporal dimension as well as adding positional encoding. These are critical enhancements for handling dynamic spatiotemporal data.

These contributions represent fundamental shifts in how spatiotemporal data can be modelled using Transformer architectures, which we believe constitutes a meaningful advance in the field.

**Conclusion**

We have reflected these answers in our revised manuscript. In response to reviewers' requests for visualizing spatiotemporal dependencies, we have also added Figures 3 and 4, showcasing model predictions across time steps for individual sensors. This provides a clearer illustration of T-Graphormer’s ability to capture spatiotemporal patterns. We hope these changes address the concerns raised and demonstrate the novelty, effectiveness, and broad applicability of T-Graphormer.

We look forward to further correspondence and are happy to provide any additional clarifications.

---

### Author Response · Authors · 2024-11-27
**Phase 2 Comment, Updates to Manuscript**

Dear Reviewers and Area Chairs,

Thank you again for your efforts during the review process. Your feedback has been very helpful in shaping our work towards better clarity and rigour.

On that note, here are some of the changes we made to improve our manuscript:
## Main Text:

* **Related Works and Baselines Comparisons**:

    We included two additional baselines and added their prediction results on the PEMS-BAY and METR-LA datasets in **Table 2**. The table has been reorganized, ranking models based on their performance on the horizon-12 prediction. Additionally, the second-best-performing model for each metric is now underlined for better readability.
* **Overall Writing**:

    Corrected typos. Improved phrasing at necessary parts (e.g. Conclusion) to ensure smoother flow and coherence.


## Appendix:
* **Additional Results**:

    Summarized the T-Graphormer prediction results. Notably, as our training loss is the mean squared error, the most significant improvement of our method over the baselines is observed in the RMSE test metric. Furthermore, the prediction improvements are more pronounced in larger datasets.

* **Figures**:

    * Added **Figures 3, 4** to visualize T-Graphormer's predictions on PEMS-BAY and METR-LA datasets.
    * Added **Figure 8** to support pre-training potentials mentioned in the conclusion.



Thank you again for your time and feedback.

We look forward to further active discussions. We are more than happy to provide clarifications or additional details.

Best,

Authors

---

### Meta-Review · Area_Chair_NcNb · 2024-12-17

**Metareview:**

This paper proposes a transformer model that aims to model spatiotemporal correlations directly for forecasting applications. In particular, it extends the Graphormer architecture to incorporate temporal dynamics. The proposed model is evaluated on some traffic prediction benchmark datasets.

Major strengths:
- This work explores a tighter integration of the spatial and temporal dimensions by operating on spatiotemporal data directly.
- The structural encoding techniques in T-Graphormer are extended to the temporal domain for spatio-temporal forecasting.

Major weaknesses:
- The technical contributions of this work are not very novel and significant.
- The original submission only included very few benchmark datasets and baseline methods for comparison, but they were slightly expanded during the rebuttal period.
- A more comprehensive literature review should be conducted, especially to include the very recent works.
- The linguistic quality and presentation of this paper have room for improvement.

We thank the authors for making an effort to address some issues raised in the reviews, which includes conducting more experiments. Nevertheless, we still feel that even the revised manuscript is below the acceptance standard of ICLR for the major weaknesses as summarized above. The authors are encouraged to improve their paper for future submission by considering the comments and suggestions of the reviewers.

**Additional Comments On Reviewer Discussion:**

All reviewers participated in the discussion after reading the author responses. Although some reviewers increased their overall ratings during the discussion period, all reviewers feel that even the revised paper is still below the acceptance standard of ICLR.

---

### Decision · Program_Chairs · 2025-01-22

Reject